# A promising perovskite primary explosive

Yongan Feng [1] ✉, Jichuan Zhang[2], Weiguo Cao [1], Jiaheng Zhang [3] ✉ & Jean'ne M. Shreeve [2] ✉

A primary explosive is an ideal chemical substance for performing ignition in military and commercial applications. For over 150 years, nearly all of the developed primary explosives have suffered from various issues, such as troublesome syntheses, high toxicity, poor stability or/and weak ignition performance. Now we report an interesting example of a primary explosive with double perovskite framework, $\{(C_6H_{14}N_2)_2[Na(NH_4)(IO_4)_6]\}_n$ (**DPPE-1**), which was synthesized using a simple green one-pot method in an aqueous solution at room temperature. **DPPE-1** is free of heavy metals, toxic organic components, and doesn't involve any explosive precursors. It exhibits good stability towards air, moisture, sunlight, and heat and has acceptable mechanical sensitivities. It affords ignition performance on par with the most powerful primary explosives reported to date. **DPPE-1** promises to meet the challenges existing with current primary explosives, and this work could trigger more extensive applications of perovskite.

Primary explosives are a class of high-energy materials that perform precise ignition or start-up in commercial, military, and space exploration applications. Over the past 150 years, countless numbers of energetic substances have been designed and screened as possible initiating explosives, including transition metal-based, potassium-based, and organics[1]. However, transition metal-based substances suffer from toxic heavy metal and virulent organic precursors[2–9], potassium-based substances have the problems of tedious synthesis (high cost) and weak ignition[10–18], while organic substances are unstable and have troublesome preparations (e.g., toxic solvents and dangerous reactions)[19–22]. It seems an impossible but attractive challenge to develop primary explosives with green, low cost, and powerful ignition performance including acceptable stability.

In response to the above challenge, an $A_2BB'X_6$-type perovskite initiating substance $\{(C_6H_{14}N_2)_2[Na(NH_4)(IO_4)_6]\}_n$ (**DPPE-1**), in which the structure is very different from those of traditional primary explosives was developed (Fig. 1). Moreover, it shows a series of advantages, such as being free of highly toxic components, simple and green synthesis, good stability towards air, moisture, sunlight, heat and mechanical stimulation, and high ignition performance, demonstrating that perovskites with reasonable design have obvious advantages in the development of advanced primary explosives.

Recently, there has been intense interest in the broad family of materials based on perovskite frameworks[23–30]. Interest in these systems derives from the confluence of low-cost solution processing, chemical and functional diversity, and tunable structure and properties[31–33]. In addition, reasonably designed perovskites can also eliminate toxicity effectively, and improve stability under ambient conditions[34–37]. Primary explosives that topologically mimic perovskites are likely to be a new generation of primary explosives. A previous studies have reported several sensitive energetic perovskites $(H_2A)[Ag(ClO_4)_3]$ and indicated that they may be potential primary explosives[38,39]. However, the lack of data on ignition performance makes it highly uncertain whether these substances can be used as primary explosives. Moreover, these silver-based perovskites cannot avoid heavy metal pollution and unaffordable costs (noble metal Ag). Therefore, we started the study of perovskite primary explosives with advantages of green, low cost and high ignition performance. The focus is on the suitable chemical composition which can magically change the properties of a perovskite giving rise to a desired ignition function. For a perovskite framework, it is reasonable to configure a monovalent periodate anion $(IO_4^-)$ at the X site, since it has both a powerful oxidizing ability to meet a desired high ignition performance as well as an acceptable sensitivity to realize reliable ignition. Our previous studies have confirmed that $IO_n^-$-based energetic substances

[1]School of Environment and Safety Engineering, North University of China, 030051 Taiyuan, China. [2]Department of Chemistry, University of Idaho, Moscow, ID 83844-2343, USA. [3]Sauvage Laboratory for Smart Materials, Harbin Institute of Technology, 518055 Shenzhen, China. ✉e-mail: fengyongan0918@126.com; zhangjiaheng@hit.edu.cn; jshreeve@uidaho.edu

(a) transition metal-based primary explosives

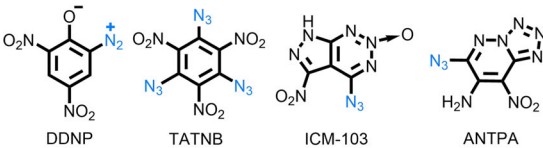

MF

LA

LS

NHP

NTCA

(b) potassium-based primary explosives

K₂DNABT

K₂DNAT

K₂BDAF

(d) **This work**

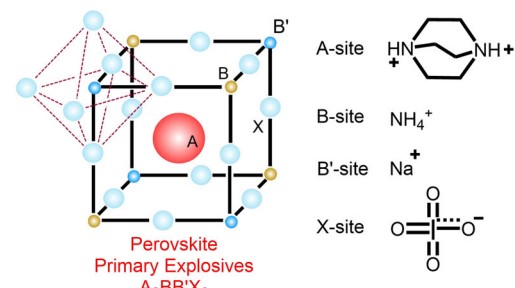

A-site

B-site NH₄⁺

B'-site Na⁺

X-site

Perovskite
Primary Explosives
A₂BB'X₆

☑ Less toxic componets and green decomposition products
☑ low-cost and green one-pot synthesis, water, r.t.
☑ good stability toward air, moisture, sunlight and heat
☑ acceptable mechanical sensitivity
☑ excellent ignition performance

(c) organic primary explosives

DDNP

TATNB

ICM-103

ANTPA

**Fig. 1 | Several types of primary explosives: traditional and our reported ones.**
**a** MF is mercury fulminate[3,4]. LA is lead azide (PbN₆)[3,4]. LS is lead styphnate
(PbC₆HN₃O₈)[3,4]. NHP is nickel hydrazine perchlorate ([Ni(N₂H₄)₃](NO₃)₂)[5]. NTCA is
5-nitrotetrazolato copper ammonium ((NH₄)₂[Cuᴵᴵ(NT)₄(H₂O)₂])[4]. **b** K₂DNABT is
dipotassium 1,1'-dinitramino-5,5'-bistetrazolate[10]. K₂DNAT is dipotassium 1,5-di(ni-
tramino)tetrazolate[12]. K₂BDAF is dipotassium 4,4'-bis (dinitromethyl)-3,3'-
azofurazanate[14]. **c** DDNP is 2-diazo-4,6-dinitrophenol[20]. TATNB is 1,3,5-triazido-
2,4,6-trinitrobenzene[4]. ICM-103 is 6-nitro-7-azido-pyrazol[3,4-d][1,2,3]triazine-2-
oxide[20]. ANTPA is 6-azido-8-nitro-tetrazolo[1,5-b]pyridazine -7-amine[21]. **d** This study
reports an energetic perovskite with the general formula A₂BB'X₆, where A is 1,4-
diazabicyclo[2.2.2]octane (or dabconium, or DABCO²⁺, or H₂dabco²⁺), B is sodium
(Na⁺), B' is ammonium (NH₄⁺), and X is periodate (IO₄⁻).

exhibit strong oxidizing, high sensitivity, and rapidly exothermic
behavior with iodine, which exhibits a bactericidal effect and is bio-
friendly as the decomposition product[39–41], thus showing great
potential as green primary explosives. The latest research also shows
the possibility of periodate-based primary explosives[42,43]. In addition,
periodate (IO₄⁻) tends to form three-dimensional frameworks through
inter-ion interactions (Supplementary Fig. 1). However, currently
reported initiating substances all belong to single-perovskite energetic
materials, while double-perovskite energetic materials have not been
reported yet, especially the existing synthesis methods of periodate-
based perovskites are uneconomical and dangerous[43], and their igni-
tion performance still needs to be improved[43]. Now, we have reported
a double-perovskite primary explosive (A₂BB'X₆ form) with excellent
ignition performance. Non-toxic cations Na⁺ and NH₄⁺ are selected to
be located at sites B and B', respectively, since both of them benefi-
cially give rise to a compact structure, enhance biocompatibility and
assist the reaction in proceeding in aqueous solution. As for the A site,
the size of the framework formed by the interconnection of Na⁺ (or
NH₄⁺) and IO₄⁻ is estimated to be ~7.42 Å × 7.42 Å × 7.42 Å[44,45]. To obtain
the maximum crystal density and filling coefficient, dabconium
(H₂dabco²⁺, C₆H₁₄N₂²⁺) with an effective radius 3.39 Å is preferred
among a series of reported organic amine cations[46]. All involved
cations and anions are common stable substances which are envir-
onmentally friendly. They are expected to be held together in aqueous
solution to form the target organic–inorganic perovskite by simple
self-assembly reaction based on interion interactions.

## Results

### Synthesis and crystalline structure
With the above considerations in mind, the synthesis of the perovskite
primary explosive was undertaken. The synthetic process of

{(C₆H₁₄N₂)₂[Na(NH₄)(IO₄)₆]}ₙ (**DPPE-1**) is very simple, green, and eco-
nomical, namely, NaIO₄, NH₄Cl and dabconium dihydrochloride
(H₂dabcoCl₂) were introduced into water and stirred at room tem-
perature, resulting in the precipitation of a large amount of white solid
within seconds. This gave rise to a granular crystalline product in good
yield (72.1 wt%) after filtration to leave a clear colorless solution (Fig. 2
and Supplementary Fig. 2). Its structure has been characterized by
elemental analysis, infrared spectrum, nuclear magnetic resonance,
and single-crystal X-ray diffraction (Supplementary Figs. 3–8).

Single-crystal X-ray diffraction determination shows that DPPE-1
crystallizes in a structure in the *cubic* space group *Pa-3* having four
formula units per unit cell (a = b = c = 14.8 Å) and a high crystal den-
sity of 2.89 g cm⁻³ at 120 K (Supplementary Table 1). With H₂dabco²⁺
as an A-site cation, NH₄⁺ as a B-site cation, Na⁺ as a B´-site cation, and
IO₄⁻ as the X-bridge, the structure of DPPE-1 can be accurately
described as an A₂BB'X₆-type double perovskite, which is different
from all those reported energetic materials with ABX₃-type single-
perovskite structure[25,47–50]. The experimental elemental analysis
shows that the composition of DPPE-1 is 10.14% for C, 2.19% for H, and
4.89% for N, respectively (Supplementary Information 3. Synthesis),
which is highly consistent with the theoretical calculation obtained
from the double-perovskite substance, confirming the formation of
DPPE-1 rather than reported (H₂dabco)[Na(IO₄)₃] (DAI-1), (H₂dabco)
[(NH₄)(IO₄)₃] (DAI-4) or (H₂dabco)[Na(H₄IO₆)₃] (DAI-X1). In terms of
composition, DPPE-1 is similar to some energetic perovskites already
reported, such as (H₂dabco)[Na(ClO₄)₃] (DAP-1), (H₂dabco)[(NH₄)
(ClO₄)₃] (DAP-4), DAI-1, DAI-4, and DAI-X1. However, it would be a
great mistake to classify them as very close analogues. In most fields,
the order of magnitude increase caused by structural changes means
more interesting properties and functions. For single perovskites,
there are theoretically $N^3$ (N = 1,2,3,……n) molecular combinations,

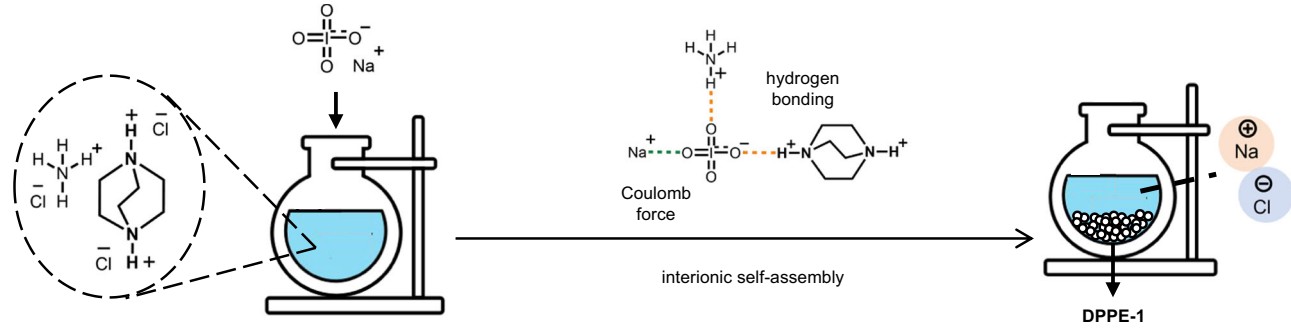

**Fig. 2 | The low-cost and green synthesis of DPPE-1.** Dabconium dihydrochloride ($H_2dabcoCl_2$, 1 M) and ammonium chloride ($NH_4Cl$, 2 M) were dissolved in water by vigorous stirring at room temperature, then the aqueous solution of sodium metaperiodate ($NaIO_4$, 6 M) was added to the above mixed solution, and the target substance DPPE-1 quickly precipitated within a few seconds (yield: >70%). The synthesis of the energetic double perovskite is a typical self-assembly process, which involves three inter-ionic interactions: hydrogen bonding between $H_2dabco^{2+}$ and $IO_4^-$, hydrogen bonding between $NH_4^+$ and $IO_4^-$, and coulomb force between $H_2dabco^{2+}$, $NH_4^+$ and $IO_4^-$.

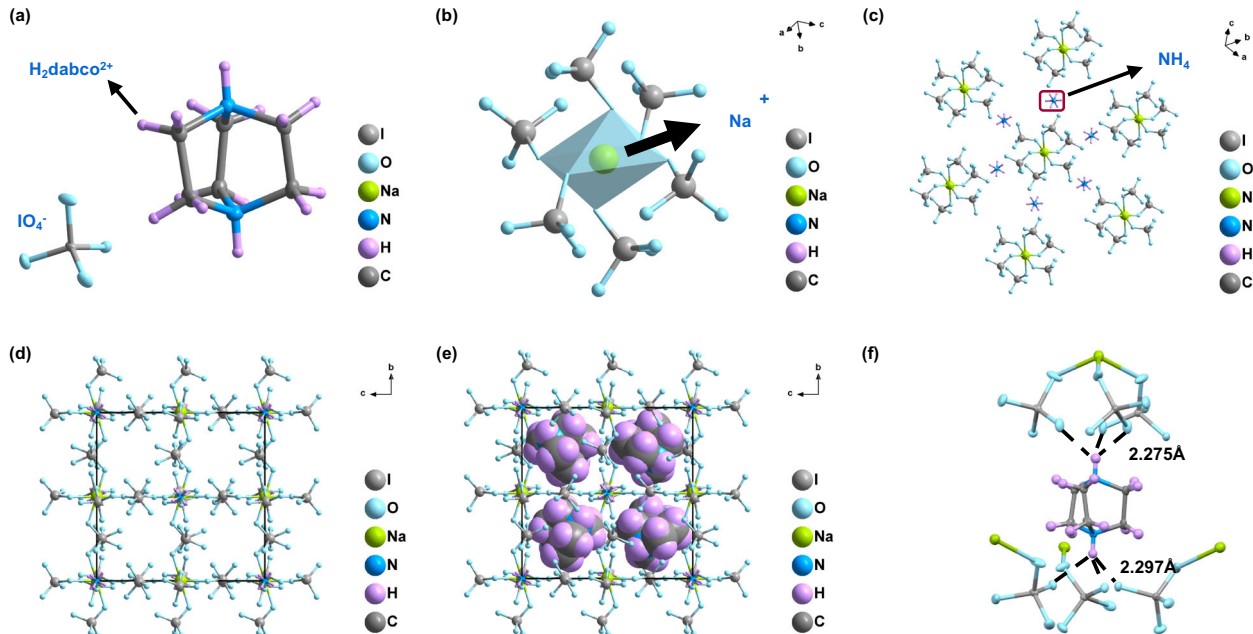

**Fig. 3 | Hierarchical self-assembly of DPPE-1 with well-defined primary, secondary, tertiary, and quaternary structures. a** Primary structures of the metaperiodate $IO_4^-$ anion and $H_2dabco^{2+}$. **b** Secondary structural unit (*Octa-small*) composed of $Na^+$ and $IO_4^-$; based on the coordination bond ($Na \cdots O$) and *Coulombic* forces ($Na+/IO_4^-$). **c** Tertiary structural unit (*Octa-large*) assembled by $NH_4^+$, for which a twisted octahedral configuration (*Octa-small*) occurs through interionic interactions (hydrogen bonding and *Coulombic* forces). **d** Unfilled cubic framework. **e** Cubic framework filled with $H_2dabco^{2+}$ cations. **f** Hydrogen bonds between the four twisted octahedral configurations (*Octa-small*) and the $H_2dabco^{2+}$ cations.

while for double perovskites, the variations could be extended to $N^4$ ($N = 1,2,3,......n$). Obviously, double perovskite energetic materials own more variations, which provides good opportunities for the discovery of some new functions and applications. Interestingly, we also noted that DAI-X1 and DAI-4 were respectively formed in the absence of ammonium ions and sodium ions, while DPPE-1 was synthesized in the presence of both ions, which demonstrated the structural and functional diversity of energetic perovskites.

**DPPE-1** exhibits a hierarchical self-assembly configuration with well-defined primary, secondary, tertiary, and quaternary structures (Fig. 3). In the unit cell, each $Na^+$ is bonded to six $IO_4^-$ ions via coordination bonds ($Na \cdots O$) and Coulombic forces ($Na^+/IO_4^-$), forming a twisted octahedral configuration (Fig. 3b, *Octa-small*), which is then connected with six adjacent $NH_4^+$ ions through hydrogen bonds (N-H$\cdots$O, 2.257 Å) and Coulombic forces ($NH_4^+/IO_4^-$) to form a regular, larger octahedral structure (Fig. 3c, *Octa-large*). As the *Octa-large* continues to expand along the three axes based on the interionic interactions, a regular cubic framework with eight cavities is eventually formed (Fig. 3d), each of which is occupied by a dabconium cation ($H_2dabco^{2+}$) (Fig. 3e). With the assistance of the hydrogen bonds (N-H$\cdots$O), each $H_2dabco^{2+}$ is completely connected to four neighboring *Octa-small* creating a unique double tetrahedral configuration (Fig. 3f), resulting in the calculated filling coefficient as high as 80.7% (Supplementary Fig. 9), which may be the highest in the field of high-energy materials to date.

Interestingly, all the observed hydrogen bonds are H$\cdots$O interactions, and most of the bond lengths are less than expected. For example, the bond lengths of N-H$\cdots$O and C-H$\cdots$O are 2.21–2.46 Å and 2.41–2.60 Å (Supplementary Table 6), whereas the expected bond length is 2.46 Å, making these hydrogen bonds relatively short from a statistical point of view[51]. We think that the synergistic effect of coulomb forces may play a positive role. Notably, both $NH_4^+$ and $H_2dabco^{2+}$ are linked to $IO_4^-$ by six N-H$\cdots$O hydrogen bonds, so none of the involved ions have extra charge and hydrogen atoms capable of forming strong attraction with water molecules, which may help explain why **DPPE-1** is able to exist as a stable material and precipitate

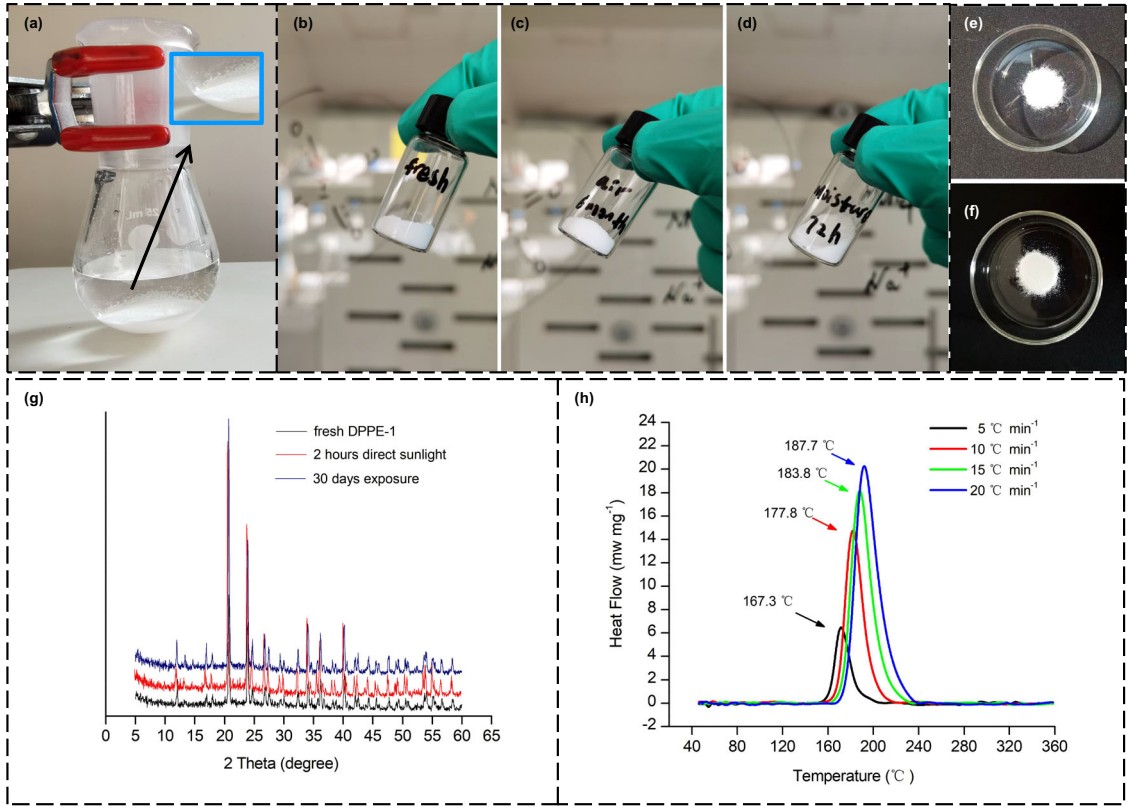

**Fig. 4 | Stability of DPPE-1. a** DPPE-1 precipitated from the solution. **b** Fresh DPPE-1. **c** Exposure to air for 6 months. **d** Exposure to moisture for several days. **e** Exposure to direct sunlight for 2 h. **f** Exposure to sunlight for 30 days. **g** Powder X-ray diffraction (PXRD) measurements in light stability tests. **h** DSC curves in thermal stability tests.

easily from aqueous solutions, thus leading to simple synthesis and separation methods. The combination of stable chemical composition, abundant interionic interactions and strengthened three-dimensional framework is also likely to lead to a stable perovskite primary explosive.

## Stability studies

To verify whether **DPPE-1** is stable under ambient conditions, its stability towards air, moisture, sunlight, heat, and mechanical stimuli was investigated, along with its long-term storage stability. Considering that **DPPE-1** is synthesized, crystallized, and filtered from an aqueous solution, we conclude that it should be stable to air and moisture. Subsequent experiments confirmed that no matter that **DPPE-1** was exposed to air for 6 months or moisture for several days, no changes in color and morphology were observed (Fig. 4a–d). The chemical stability of **DPPE-1** was further verified by light stability test (Fig. 4e, f), which showed that **DPPE-1** remained in its original state after 2 h of direct sunlight and 30 days of sunlight exposure under ambient conditions, which was confirmed by powder X-ray diffraction analysis (Fig. 4g).

Differential scanning calorimetry (DSC) was used to assess the stability towards heat loading (details see Supplementary Information). Its thermal decomposition occurs with an onset temperature at 161.3 °C (Fig. 4h), comparable to dinitrodiazophenol (DDNP, 157.0 °C), 6-nitro-7-azido-pyrazol[3,4-d][1,2,3]triazine-2-oxide (ICM-103, 160.3 °C), and 6-azido-8-nitro-tetrazolo [1,5-b]pyridazine-7-amine (ANTPA, 163.0 °C)[19,20], which satisfies most military and civilian needs. In addition, its thermal decomposition temperature is lower than that of reported perchlorate-based perovskite energetic materials (perchlorate-based, 179–371 °C)[25,47–50], but it is comparable to four reported periodate-based analogues (152–169 °C)[42,43]. It should be noted that although **DPPE-1** meets the minimum thermal decomposition

temperature requirements ($T_{dec} \geq 150$ °C) of green primary explosives, it is more attractive to develop perovskite ignition materials with higher heat resistance (e.g. $T_{dec} \geq 180$°C or even $\geq 200$ °C)[4,10,52,53]. The long-term storage stability test was further performed by storing **DPPE-1** at atmospheric pressure and 75 °C for 48 h (Supplementary Table 9). The results showed that the mass loss of the **DPPE-1** was almost negligible (≤0.05%), suggesting excellent long-term storage stability.

In addition, we determined the mechanical sensitivity of **DPPE-1** (Supplementary Table 10). As evident from Table 1, its impact sensitivity (IS) and friction sensitivity (FS) are 3.5 J and 5.0 N, respectively. The impact sensitivity is comparable to those reported for most primary explosives, and the friction sensitivity is better than that of widely used military primary explosive LA[12] and close to those reported periodate-based single perovskites used as biocidal agents (<5 N)[42]. Accordingly, it has acceptable mechanical sensitivity. The data in Table 1 also show that the oxygen balance ($\Omega_{CO} = -4.52\%$) of **DPPE-1** is higher than those of other initiating substances–possibly the highest oxygen balance for a primary explosive to date.

## Ignition performance

The minimum primary charge (MPC) is the most important parameter for evaluating the ignition performance of a primary explosive. In this study, MPC was determined with the device shown in Fig. 5a–c (details see Supplementary Information 10. Ignition performance). The lead plate was successfully penetrated when the loading weight of DPPE-1 was 20 mg and 10 mg (Fig. 5d, e). We further reduced the loading weight to 5 mg, which is considered as the ultimate weight, because the surface of RDX cannot be completely covered if the amount of **DPPE-1** is less than 5 mg. Interestingly, 5 mg of **DPPE-1** is also able to initiate RDX reliably, making it an efficient initiating substance (Fig. 5f). It is clear that the ignition performance of **DPPE-1** is far superior to those of

**Table 1 | Physical and energetic properties of several typical primary explosives and DPPE-1**

| Items | Formula | $M^a$ (g mol$^{-1}$) | $\Omega_{CO}{}^b$ (%) | $\rho^c$ (g cm$^{-3}$) | IS$^d$ (J) | FS$^e$ (N) | $T_{dec}{}^f$ (°C) |
|---|---|---|---|---|---|---|---|
| **DPPE-1** | $C_{12}H_{32}I_6N_5NaO_{24}$ | 1414.8 | −4.52 | 2.88 | 3.5 | 5.0 | 161.3 |
| LA$^g$ | $PbN_6$ | 291.3 | −5.49 | 4.80 | 2.5–4 | 0.1–1.0 | 315.0 |
| SA$^h$ | $AgN_3$ | 149.9 | −21.35 | 4.81 | >2.5 | >0.1 | >297.0 |
| CA$^i$ | $CuN_6$ | 147.6 | −10.84 | 2.2-2.58 | ≤1.0 | ≤0.1 | 205.0 |
| K$_2$DNABT$^j$ | $C_2K_2N_{12}O_4$ | 291.3 | 0 | 2.11 | 1.0 | ≤1.0 | 200.0 |
| K$_2$DNAT$^k$ | $CK_2N_8O_4$ | 266.3 | −6.02 | 2.18 | 1.0 | <5.0 | 240.0 |
| K$_2$BDAF$^l$ | $C_6HK_2N_{10}O_{10}$ | 450.3 | +10.66 | 2.04 | 2.0 | 20.0 | 229.0 |
| DDNP$^m$ | $C_6H_2N_4O_5$ | 210.1 | −15.23 | 1.72 | 1.0 | 24.7 | 157.0 |
| ICM-103$^n$ | $C_4HN_9O_3$ | 223.1 | −10.75 | 1.86 | 4.0 | 60.0 | 160.3 |
| ANTPA$^o$ | $C_4H_2N_{10}O_2$ | 222.1 | −21.61 | 1.82 | 5.0 | 120.0 | 163.0 |

*LA* lead azide, *SA* isilver azide. *CA* copper azide. See Fig. 1 for the chemical structures of other substances.
$^a$formula weight. $^b$oxygen balance. $^c$crystal density. $^d$impact sensitivity. $^e$friction sensitivity. $^f$decomposition temperature, $^g$ref. 10, $^h$ref. 1, $^i$ref. 1 & 7, $^j$ref. 10, $^k$ref. 12, $^l$ref. 14, $^m$ref. 20, $^n$ref. 20, $^o$ref. 21.

recently reported green primary explosives with claimed initiation efficiency (e.g. DDNP, ICM-103, ANTPA, K$_2$DNABT, K$_2$DNAT), and is comparable to those of most powerful primary explosives (e.g. PbN$_6$, AgN$_3$, CuN$_6$) (Supplementary Table 11) and periodate-based perovskites[43]. However, **DPPE-1** can't be a perfect replacement for PbN$_6$ unless its thermal decomposition temperature is higher than 200 °C. As for K$_2$DNABT, K$_2$DNAT, and K$_2$BDAF, they may be high-performance primary explosives, but the lengthy manufacturing process makes them almost impossible to replace PbN$_6$. In industry, the excellent initiation performance of **DPPE-1** would enable excellent economic, social, and environmental benefits, contributing to a significant reduction in primary explosive production and lowering both environmental hazards and casualties.

How to explain the mechanism of such excellent ignition performance? Here, we calculate the heat of formation of **DPPE-1**, and then evaluate its energy level by using program EXPLO7.0 (Supplementary Table 12). The results show that its detonation velocity (D) and detonation pressure (P) are <5500 m/s and <18.5 GPa, respectively, which are lower than those of most primary explosives, indicating that the impressive initiation performance of **DPPE-1** is hard to be well explained from the energy level. So, we turned our attention to oxidation of IO$_4^-$. The ignition performance of the primary explosive is generally considered to depend mainly on the oxidation of the oxidizing components. For example, energetic substances based on perchlorate (ClO$_4^-$) are usually used to construct high-performance primary explosives[54,55], with the reason that ClO$_4^-$ has stronger oxidation than nitrite (NO$_3^-$), chlorate (ClO$_3^-$), sulfate (SO$_4^{2-}$), styphnate (C$_6$HN$_3$O$_8^{2-}$), and so on, thus bringing excellent ignition performance. According to the Tables of Standard Electrode Potentials[56], however, the standard electrode potential ($E_O$) of IO$_4^-$ is 1.314 V, while that of ClO$_4^-$ is 1.389 V. Obviously, the oxidation of ClO$_4^-$ is stronger than that of IO$_4^-$, which demonstrates that the good initiation performance of **DPPE-1** may be related to some unknown influencing factors besides the strong oxidation of IO$_4^-$. We think that molecular stability is likely to play a crucial role, so we compare the Gibbs Free Energy ($\Delta G$) of (H$_2$dabco)$_2$[Na(NH$_4$)(IO$_4$)$_6$]$_n$ and (H$_2$dabco)$_2$[Na(NH$_4$) (ClO$_4$)$_6$]$_n$ (Supplementary Table 13). The results show that the $\Delta G$ of (H$_2$dabco)$_2$[Na(NH$_4$)(IO$_4$)$_6$] ($\Delta G = -144.95$ kcal/mol) is higher than that of (H$_2$dabco)$_2$[Na(NH$_4$)(ClO$_4$)$_6$] ($\Delta G = -201.98$ kcal/mol), that is, the ClO$_4^-$ in (H$_2$dabco)$_2$[Na(NH$_4$)(ClO$_4$)$_6$] has a good stabilizing effect, while the material based on IO$_4^-$ is relatively unstable, which will help us to understand the high mechanical sensitivity and strong initiation performance of **DPPE-1**. In any case, there is no doubt that halogens play a decisive role in the initiation performance of energetic perovskites according to our available studies. For example, compared with ClO$_4^-$-based materials, IO$_4^-$-based materials are more likely to exhibit high sensitivity, rapid exothermic processes, and strong initiation performance, while BrO$_4^-$-based materials are generally too unstable to be synthesize and used as energetic materials.

Furthermore, the explosive products of the perovskite primary explosive are discussed in detail through theory and experiment. It is well known that reducing the toxicity of explosive products is the main driving force to promote the development of green primary explosives. The rise of green primary explosive stems from the elimination of heavy metal toxicity from solid decomposition products (e.g. Pb and Hg) of early primary explosives[53]. Theoretically, highly sensitive substances without heavy metals Pb and Hg can be regarded as potential green primary explosives. This is an important reason why some substances are classified as green primary explosives even though they contain highly toxic components, including silver azide (SA), nickel hydrazine nitrate (NHN), copper(I) 5-nitrotetrazolate (DBX-1), bis-(5-nitrotetrazole)tetraamine cobalt(III) perchlorate (BNCP), 2-diazo-4,6-dinitrophenol (DDNP), cyanuric triazide (CTA), potassium 4,6-dinitro-7-hydroxybenzofuroxan (KDNP), and potassium 4,6-dinitrobenzofuroxan (KDNBF)[53]. Although the perchlorate-based primary explosives have reduced toxicity, their gaseous explosive product (e.g. HCl) is still uncomfortable. Component IO$_4^-$ is also biotoxic, however, its toxicity is at most comparable to the components such as hydrazine, azide, nitrotetrazolate, nitrophenol, benzofuroxan, and cyanuric triazide in the above-mentioned green primary explosives, and the main solid product (I$_2$) of periodate-based primary explosives is a typical material with sterilization and disinfection functions. Therefore, **DPPE-1** should be considered as a non-toxic design ideas according to the current creteria. Energetic biocidal agents were put forward based on similar ideas[41,42]. Calculations based on the EXPLO7.0 program show that most of detonation products of **DPPE-1** are non-toxic and less toxic, with a mass percentage of I$_2$ of 51.7% (Supplementary Table 14). To confirm the existence of I$_2$ in the detonation product, we filled the **DPPE-1** in a pressure-resistant glass bottle and heated it in an oven to 200 °C As a result, we heard a huge explosion and the bottle was completely broken (Supplementary Fig. 12a, b); then we filled **DPPE-1** in a Teflon reactor and heated it in an oven to 200 °C. As a result, no explosion was heard and the detonation product stained the inner wall of the container purple, indicating the possible formation of I$_2$ (Supplementary Fig. 12c, d). Further tests showed that the aqueous solution of the purple substance was yellow and immediately formed a blue solution when mixed with starch (Supplementary Fig. 12e−g), confirming the presence of a large amount of I$_2$. So IO$_4^-$ is suitable for the construction of green primary explosives.

## Discussion

An interesting primary explosive (**DPPE-1**) has been reported, which consists of an organic−inorganic double perovskite structure reasonably different from the structure of traditional primary explosives. Its

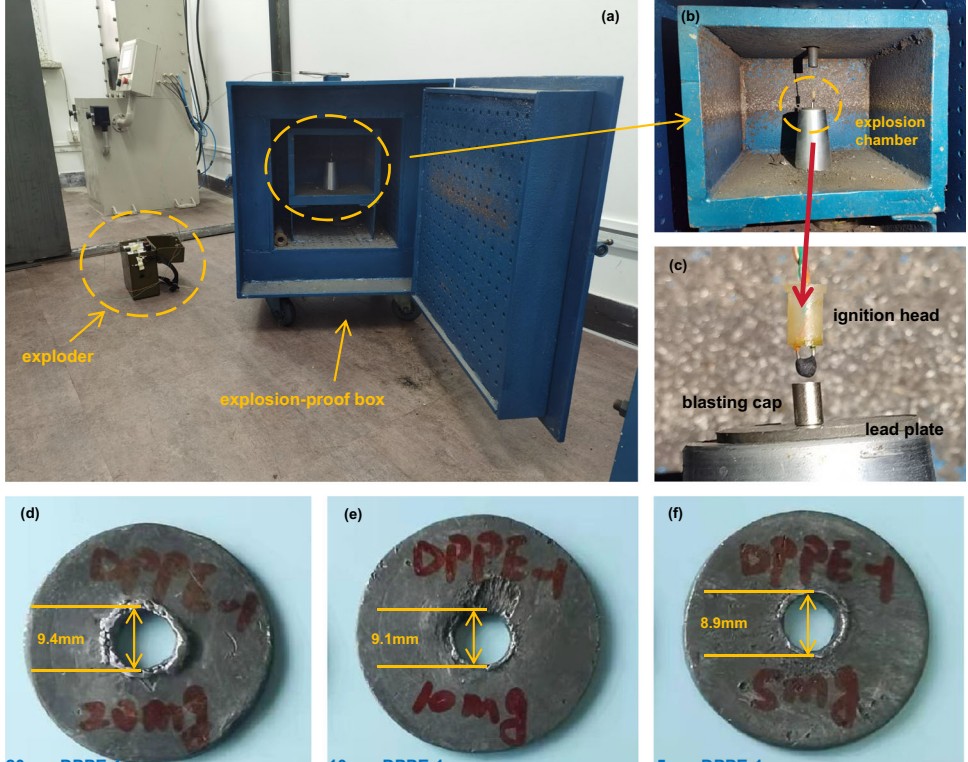

**Fig. 5 | Evaluation of the ignition performance of the primary explosive DPPE-1.** **a** the ignition performance test system. The exploder is a portable power supply, which is used to ignite the ignition head. **b** Explosion chamber. It is a metal box used to provide safety protection. **c** The combination of ignition head, detonator, and lead plate in the testing device. It is a key part for testing ignition performance. **d–f** Test results for the minimum primary charge (MPC).

synthesis involves a simple green one-pot process and most of the decomposition products are non-toxic or less toxic, which makes this primary explosive both cost-effective and environmentally friendly. Single-crystal X-ray diffraction confirms its cubic structure to be similar to the well-known organic–inorganic hybrid perovskites formed through self-assembly of hierarchical structures. Detailed studies show that the primary explosive **DPPE-1** has an excellent comprehensive performance, such as high oxygen balance (−4.52%), high crystal density (2.88 g cm$^{-3}$), high filling factor (80.7%), good environmental tolerance (to air, moisture, and light), reasonable thermal stability ($T_{dec}$ = 161.3 °C), and acceptable mechanical sensitivity (IS = 3.5 J; FS = 5.0 N). Most impressive is its ultra-high initiation performance (MPC ≤ 5 mg). These factors led to the discovery of perovskite materials with ignition function, demonstrating that this organic–inorganic perovskite is an exceptional platform for developing advanced primary explosives. The search for perovskite-type green primary explosives with thermal decomposition temperatures higher than 200 °C or even 250 °C will be the focus of priority consideration in the future. Given the undoubted importance of perovskites and primary explosives in various critical engineering applications, the discovery of **DPPE-1** is likely to be a pioneer in materials science and engineering technology. It is expected that additional perovskite primary explosives or highly energetic perovskites will continue to be produced through independent structures or systematic combinations of organic and inorganic components.

## Methods
### Safety precautions
**DPPE-1** is a highly explosive, sensitive material, and it should be handled with extreme caution using proper safety equipment, such as protective gloves and coats, a face shield, and an explosion-proof baffle.

### Materials
Sodium metaperiodate (99.5%), ammonium chloride (99.8%), and triethylene diamine (98%) were purchased from Shanghai Aladdin Biochemical Technology Co., Ltd. Dabconium dihydrochloride was obtained by reacting triethylene diamine with dilute hydrochloric acid.

### Characterization
Infrared spectra (IR) were recorded on a Bruker Equinox 55 infrared spectrometer. Elemental analysis (C, H, and N) was performed on a varioMICRO cube fully automatic trace element analyzer. $^{1}$H and $^{13}$C NMR spectra were recorded on a Bruker Advance 600 nuclear magnetic resonance spectrometer. Powder X-ray diffraction (PXRD) measurements were performed on a Bruker D8 advance diffractometer. The single-crystal X-ray diffraction data were collected on a Rigaku AFC-10/Saturn 724 + CCD diffractometer. Thermal decomposition temperatures were determined using differential scanning calorimetry (DSC) on a CDR-4 from Shanghai Precision & Scientific Instrument Co. Ltd. The long-term storage stability and minimum primary charge (or initiation performance test) were measured according to the method given by GJB 5891-2006. The impact and friction sensitivity measurements were performed using a standard BAM Fall hammer and a BAM friction tester.

### Synthesis of DPPE-1
Dabconium dihydrochloride (0.37 g, 2 mmol) and ammonium chloride (0.0535 g, 1 mmol) were dissolved in 5 mL water by vigorous agitation (600 r min$^{-1}$) at room temperature. Subsequent addition of 8 mL sodium metaperiodate (NaIO$_4$, 1.28 g, 6 mmol) solution into the above mixture resulted in the immediate precipitation of a white solid and the reaction solution became clear and colorless within 2–3 s. The resulting precipitate was filtered, washed with an ice/water mixture (2 × 3 mL), and then sequentially dried under sunlight and vacuum to

yield the target compound **DPPE-1** as a colorless solid. Yield: 1.02 g, 72.1%. DSC (5 °C min$^{-1}$, °C): 161.3 (dec.); IR (KBr pellet, cm$^{-1}$): $\tilde{v}$ 3122 (m), 3034 (w), 1475 (m), 1419 (s), 1328 (w), 1214 (m), 1056 (s), 830 (s). $^1$H NMR (600 MHz, DMSO-d$_6$, 25 °C): $\delta$ = 7.06 ppm (1H, NH), 3.36 (2H, CH$_2$); $^{13}$C NMR (600 MHz, DMSO-d$_6$, 25 °C): $\delta$ = 43.90 ppm; EA calculated for C$_{12}$H$_{32}$I$_6$N$_5$NaO$_{24}$ (1414.82 g mol$^{-1}$): C 10.19, H 2.28, N 4.95; Found: C 10.14, H 2.19, N 4.89.

## Data availability

Data that support the findings of this study are available from the corresponding authors on request. The supplementary crystallographic data generated in this study have been deposited in the Cambridge Crystallographic Data Centre (CCDC) database under accession code 2173977 via www.ccdc.cam.ac.uk/data_request/cif.

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

## Acknowledgements

Financial support is acknowledged from the National Natural Science Foundation of China 22275169 (Y.F.), 21905258 (Y.F.), 21905069 (Jiaheng Zhang), 22208073 (Jiaheng Zhang), U21A20307 (Jiaheng Zhang); Shenzhen Science and Technology Innovation Committee ZDSYS20190902093220279 (Jiaheng Zhang), KQTD20170809110344233(Jiaheng Zhang), GXWD20201,230155427003-20200821181245001 (Jiaheng Zhang), GXWD20201230155427003-20200821181809001 (Jiaheng Zhang), ZX20200151) (Jiaheng Zhang), Department of Science and Technology of Guangdong Province (Grant No. 2020A1515110879 (Jiaheng Zhang). Fluorine-19 Fund (J.M.S.).

## Author contributions

Y.F., Jichuan Zhang, and J.M.S. designed the study. Y.F., Jiaheng Zhang, and J.M.S. guided the project. Y.F. carried out the synthesis, structural characterizations, ignition performance measurement and mechanism research. Jichuan Zhang performed filling coefficient calculations, stability investigation and mechanism research. W.C. performed the crystallographic structural analysis. Y.F. contributed to the schematics and photographs. Y.F., Jichuan Zhang, Jiaheng Zhang and J.M.S. co-wrote the paper. All authors discussed the results and commented on the manuscript. Y.F. and Jichuan Zhang contributed equally to this work.

## Competing interests

The authors declare no competing interests.
