## [Peer Review File · Nature Communications]

Reviewers' Comments:

Reviewer #1:

Remarks to the Author:

I have carefully read the revised paper and the authors' response to the last submission and believe that this manuscript is not suitable for publication in *Nature Communications* for three main reasons:

(1) Structure: DPPE-1 synthesized in this work is too similar to the materials already reported. The use of XO_4^- as the main component to construct energetic perovskites has been extensively studied, and the relevant literatures have been reflected in previous comments of two reviewers.

Although the authors emphasize that this structure is double perovskite $A_2BB'X_6$ with the molecular formula $(H_2DABCO)_2[Na(NH_4)(IO_4)_6]$, that is, B and B' are Na^+ and NH_4^+ respectively. But actually, according to the single crystal structure, Na^+ is formed as a whole with IO_4^- as $[I_6NaO_{24}]^{5-}$, and is not a separate B site, the author's description of double perovskite is not rigorous enough. However, even if this is a double perovskite, there have been many similar structures with different A or B or C sites reported, so this structure may be not a highlight.

(2) Ignition performance: The authors claim that this perovskite material has the best ignition performance, but as reviewer 2 said, its performance does not surpass some existing primary explosives, such as copper azide and silver nitrotetrazolium, so the authors' claim of the best performance is not appropriate. I also agree with reviewer 1 that the mechanism of such excellent ignition performance is lacking, although in the response that it was claimed to be caused by the high oxidation of IO_4^- , but this is not enough. It is almost common sense that the XO_4^- series ions can be used to construct primary explosives in the field of explosives. And now, XO_4^- is not suggested to build explosives, because these halogens are inherently more toxic, which is contrary to the authors' proposed non-toxic design ideas (Scheme 1).

(3) Thermal stability: The thermal decomposition temperature of DPPE-1 is not outstanding, although it has similar thermal stability with the famous DDNP. But with the increasing safety requirements in the field of explosives, it was required that the thermal decomposition temperature of explosives should reach 180 degree. It is true that the current thermal stability has indeed reached the basic requirements for practical applications, but 161.3 degree is really not sufficient.

Reviewer #2:

Remarks to the Author:

Congratulations again to the outstanding and impressive new perovskite compound. It is really simple and could be of potential use.

I totally recommend publication of this manuscript in the present journal. Some of the overdrawn formulations have been changed from the previous version. This is gratefully acknowledged.

I still have a strong problem with table 1 line mpc. The values given here for the other compounds are simply not correct. All of the references given here are not determining the mpc, they only give examples or amounts of a specific detonator setup. So either you have to determine the mpc of the other compounds in your setup, with all desired values such as pressure, grain size etc. or this line should be deleted. At least the values of the compared explosives must be deleted.

I do not want to say your perovskite is bad and lower than others but the authors should know that one cannot compare reference data using different setups and PETN and RDX charges for comparison. In our preliminary tests we achieved a similar initiation capability to LA which is quite nice.

To summarize, this work should be published and is of great interest to other researchers in high energy materials research.

We have checked the synthesis, which worked quite fine. The analytic data largely fit to our

observations. Our DTA was a bit lower for example. The methodologies used by the authors are appropriate and meet the desired standards. Good luck for the future. I know there are other Chinese research groups working on the exact same class of compounds.

Reviewer #3:

Remarks to the Author:

In this manuscript, Feng et al. claimed that they find a new energetic double perovskite (named DPPE-1) could be a promising primary explosive. In particular, they reported a very low minimum primary charge (MPC) of 5 mg for DPPE-1. The present findings seems very impressive, but in my opinion, some conclusions are suspicious. Moreover, in considering that some very close analogues of energetic perovskite class have been reported in refs. 24, 37, 41, and 45-49, the novelties of DPPE-1 (neither in structure nor the usage as primary explosive) is not high enough for publication on Nature Communication.

(1) The space group is Pa-3 in main text (and cif) but Pcab in Table S1, which one should be right? As the authors had not supplied any structure factors, the full set of tests cannot be run (see the first line in checkCIF/PLATON report). The authors should upload a completed cif file embedding HKL reflections and refinement details for fully checkcif. In addition, according to the EA result, the present single-crystal structure refinement may be wrong in the site occupancy of B-site components.

(2) Page 9, line 2, "EA calculated for C₁₂H₃₂I₆N₅NaO₂₄ (1414.82 g mol⁻¹): C 10.19, H 2.28, N 4.95; Found: C 10.30, H 2.69, N 5.83." The experiment wt% value (5.83) for N element is significantly higher than the expected one (4.95). As disclosed in the ref. 41, NH₄⁺ could easily assemble into (H₂dabco)(NH₄)(IO₄)₃ (named DAI-4) with 75% yield, whereas sodium ion is relatively smaller for directly constructing (H₂dabco)[Na(IO₄)₃] (named DAI-1) from solution but giving (H₂dabco)[Na(H₄IO₆)₃] (DAI-X1) that could converted DAI-1 after dehydration at 80 °C. These facts in ref. 41, together with the mismatch in the present EA results, strongly imply that the sample used in the present manuscript is most-likely a mixture (main component should be DAI-4 with 5.96 wt% of N element, rather than the ideal DPPE-1 with 4.95 wt% of N element) or a solid solution (i.e., a larger amount of sodium sites are replaced by NH₄⁺ in the ideal DPPE-1) but not the declared pure DPPE-1. Regardless of the situations, it means that the obtained sample is more close to the known DAI-4, or the ideal DPPE-1 cannot be prepared as reliably as DAI-4, bringing troubles to subsequent applications. As all these relevant compounds have very close simulated PXRD patterns, the identify for the present sample should be very carefully (a pawley refinement maybe helpful in a certain degree).

(3) For MPC test, the present lead-plate experiment is relatively preliminary. There are many factors highly affect MPC, such as the sample purity, crystalline form, crystal quality, crystal size and morphology. Therefore the reported MPC in different literature usually fluctuate within a certain range. In addition, the practical primary charge is not only determined by MIC, but also closely related to the detonation reliability under different conditions. For instance, different from the MPC of 30 mg for well-known LA reported in ref. 10, the MPC of 7 mg for LA is usually found in some textbooks, and the practical primary charge of 30 mg LA is usually needed in order to achieve stable detonation every time. Moreover, in order to compare the initiation ability of different initiating explosives, comparative experiments should be conducted under the same conditions. In addition, in considering that the present DPPE-1 sample may be a mixture or solid solution which do not easily and reliably prepared, I strongly suggest the author to study the potential as primary explosive for DAI-4.

(4) The writing is not good and rigorous. Many author names are wrong in reference lists.

Reviewer #4:

Remarks to the Author:

The authors report a perovskite based promising Primary explosive. The work is interesting and the authors have done a systematic experimental study to address the energetic performance. However, I still feel that the Chemistry part of the manuscript is still not upto the level of Nature Communications. The major draw back of the paper is that there is no theory part to complement the experiments to make the story complete.

1. The studied compound (perovskite) is already well established and this is not the first time reported in the literature. Too much emphasis on this is not a good idea.

2. I want the authors to compare the performance of the present compound with some other Azide such as NH_4N_3 , AgN_3 , CuN_6 etc. This will be very much useful for the readers. The authors should also consider Potassium 1,1'-dinitroamino-5,5'-bistetrazolate (K2DNABT) for comparison. Please see the following paper {J. Chem. Phys. 143, 064508 (2015)}. I suggest the authors to include the above paper in the introduction part.

3. I would like to know the role of Halogen towards the energetic properties in this compound?

4. The Halogens are toxic and I would like to know how the present compound will satisfy the green energetic materials criteria.

5. Can this compound be a perfect replacement for PbN_6 if so how superior it is when compared to K2DNABT. I want the authors to compare the same. Also K2BDAF should also be compared.

As of now the comparison is not adequate for the energetic properties and I suggest the authors to revise the manuscript accordingly. Also the Hydrogen bonding part should be well discussed. The authors just present without much of discussion.

I would like to review the paper again.

A Point by Point Response to the Reviewer's Comments

Response to Reviewer #1

I have carefully read the revised paper and the authors' response to the last submission and believe that this manuscript is not suitable for publication in Nature Communications for three main reasons:

Comment 1: *"Structure: DPPE-1 synthesized in this work is too similar to the materials already reported. The use of XO_4^- as the main component to construct energetic perovskites has been extensively studied, and the relevant literatures have been reflected in previous comments of two reviewers. Although the authors emphasize that this structure is double perovskite $A_2BB'X_6$ with the molecular formula $(H_2DABCO)_2[Na(NH_4)(IO_4)_6]$, that is, B and B' are Na^+ and NH_4^+ respectively. But actually, according to the single crystal structure, Na^+ is formed as a whole with IO_4^- as $[I_6NaO_{24}]^{5-}$, and is not a separate B site, the author's description of double perovskite is not rigorous enough. However, even if this is a double perovskite, there have been many similar structures with different A or B or C sites reported, so this structure may be not a highlight."*

Response: We do not agree with your statement for the following reasons. (1) **The structure we report is new in the field of energetic materials.** Based on our knowledge, many perovskite-based energetic materials have been reported so far, but they show single-perovskite structure, while double-perovskite energetic materials have not been reported. Therefore, your statement of "DPPE-1..... is too similar to the materials already reported" is incorrect. We don't believe that it is necessary to explain this further; (2) **Has the energetic perovskite based on component IO_4^- been widely studied?** According to the related literature (*Energetic Materials Frontiers*, 2020, 1, 123-135; *Inorg. Chem.*, 2022, 61, 4143-4149; *Sci China Mater.*, 2023, 66, 1641-1648), only one paper has reported the perovskite energetic material based on IO_4^- , and it is a single perovskite material used as a biocidal agent (*Sci China Mater.*, 2023, 66, 1641-1648). The so-called "The use of XO_4^- as the main component to construct energetic perovskites has been extensively studied" is not the case. (3) **Isn't $(H_2DABCO)_2[Na(NH_4)(IO_4)_6]$ reported in our manuscript a double-perovskite energetic material?** In the published papers, some ABX_3 -type single-perovskite energetic materials, such as $(H_2dabco)[Na(ClO_4)_3]$, $(H_2dabco)[K(ClO_4)_3]$, $(H_2dabco)[Rb(ClO_4)_3]$, $(H_2dabco)[Na(IO_4)_3]$, $(H_2dabco)[K(IO_4)_3]$ and $(H_2dabco)[Rb(IO_4)_3]$, all of them use alkali metal ions as an independent B component. If you believe that " Na^+ is formed as a whole with IO_4^- as $[I_6NaO_{24}]^{5-}$ ", then none of the above substances can be regarded as perovskite energetic materials. Obviously, it isn't true. (4) **Is a substance with a certain similarity in chemical composition to other substances not innovative enough in terms of its new structure?** Energetic materials scientists should be aware that perchlorate-based perovskite materials are essentially double salts of two perchlorates. For example, $(H_2dabco)[Na(ClO_4)_3]$ is essentially composed of $H_2dabco(ClO_4)_2$ and $NaClO_4$. According to your logic, $(H_2dabco)[Na(ClO_4)_3]$ contains very similar components to $H_2dabco(ClO_4)_2$ and $NaClO_4$, so it does not exhibit a novel enough structure, and the perovskite energetic materials reported by Professor Xiaoming Chen are meaningless (*Sci. China. Mater.* 2018, 61, 1123-1128). This is unlikely, also.

In response to your above question, we have made a detailed description on the novel structure

in the revised manuscript (Page 4, lines 11-21).

Comment 2: "Ignition performance: The authors claim that this perovskite material has the best ignition performance, but as reviewer 2 said, its performance does not surpass some existing primary explosives, such as copper azide and silver nitrotetrazolium, so the authors' claim of the best performance is not appropriate. I also agree with reviewer 1 that the mechanism of such excellent ignition performance is lacking, although in the response that it was claimed to be caused by the high oxidation of IO_4^- , but this is not enough. It is almost common sense that the XO_4^- series ions can be used to construct primary explosives in the field of explosives. And now, XO_4^- is not suggested to build explosives, because these halogens are inherently more toxic, which is contrary to the authors' proposed non-toxic design ideas (Scheme 1)."

Response: We agree with part of your statement. (1) **Is it claimed in the revised manuscript that this perovskite material has the best ignition performance?** In fact, we didn't suggest such an inappropriate description in the first revised manuscript. In the Abstract, it is stated that "Impressively, it affords ignition performance on par with the most powerful primary explosives reported to date", instead of "this perovskite material has the best ignition performance,, its performance does not surpass some existing primary explosives, such as copper azide and silver nitrotetrazolium". We sincerely hope that you can reconsider your comment and notice one of our important innovations, that is, discover the ignition performance of perovskite materials, and then make a positive evaluation of our research and efforts. (2) **Why does DPPE-1 have such excellent ignition performance?** In the first revised manuscript, we explained that the ignition performance of DPPE-1 mainly depends on the strong oxidation ability of IO_4^- . However, no detailed supporting information is given. Here, we calculate the heat of formation of DPPE-1, and evaluate the detonation performance of DPPE-1 by using program EXPLO 7.0 (see Table R1). The results show that its detonation velocity (D) and detonation pressure (P) are < 5500 m/s and < 18.5 GPa, respectively, which are lower than those of most primary explosives, indicating that the impressive initiation performance of DPPE-1 is hard to be well explained from the energy level. So, we turned our attention to oxidation capacity. According to the Tables of Standard Electrode Potentials (*G. Milazzo et al 1978 J. Electrochem. Soc. 125 261C*), the standard electrode potential (E_0) of IO_4^- is 1.314V, while that of ClO_4^- is 1.389V. Obviously, the oxidation of ClO_4^- is stronger than that of IO_4^- , which shows that the good initiation performance of DPPE-1 may be related to some unknown influencing factors besides the strong oxidation of IO_4^- . We think that molecular stability is likely to play a crucial role, so we compare the Gibbs Free Energy (ΔG) of $(\text{H}_2\text{dabco})_2[\text{Na}(\text{NH}_4)(\text{IO}_4)_6]_n$ and $(\text{H}_2\text{dabco})_2[\text{Na}(\text{NH}_4)(\text{ClO}_4)_6]_n$ (see Table R2). The results show that the ΔG of $(\text{H}_2\text{dabco})_2[\text{Na}(\text{NH}_4)(\text{IO}_4)_6]$ ($\Delta G = -144.95$ kcal/mol) is higher than that of $(\text{H}_2\text{dabco})_2[\text{Na}(\text{NH}_4)(\text{ClO}_4)_6]$ ($\Delta G = -201.98$ kcal/mol), that is, the ClO_4^- in $(\text{H}_2\text{dabco})_2[\text{Na}(\text{NH}_4)(\text{ClO}_4)_6]$ has a good stabilizing effect, while the material based on IO_4^- is relatively unstable, which may help us to understand the high mechanical sensitivity and strong initiation performance of DPPE-1. Such in-depth understanding is inseparable from the continuous support of reviewers, and we are very grateful for your kindness and patience. (3) **Can IO_4^- be used to build a primary explosive?** When scientists are engaged in research, it is crucial to clarify the historical reasons behind it. The rise of green primary explosives stems from the the elimination of heavy metal biotoxicity from solid explosive products (e.g., Pb) of lead-based primary explosives.

Although the perchlorate-based primary explosives developed in recent years have reduced toxicity to some extent, its gaseous explosive product (e.g. HCl) is still uncomfortable. Therefore, we need to understand that the toxicity of products is the fundamental driving force to promote the development of green primary explosives. This is common sense known to the experts of primary explosives. Although IO_4^- has biological toxicity as the reviewer said, the main solid product of periodate-based primary explosives (I_2 , the confirmation of iodine in the product is shown in Figure R1 and Table R3) is a typical substance with sterilization and disinfection functions. Therefore, DPPE-1 should be considered as a non-toxic design idea. Energetic biocidal agents were put forward based on similar ideas (see reference “*New Promises from an Old Friend: Iodine-Rich Compounds as Prospective Energetic Biocidal Agents*” (Acc. Chem. Res. 2021, 54, 2, 332-343), and “*Periodate-based molecular perovskites as promising energetic biocidal agents*” (Sci China Mater., 2023, 66, 1641-1648)). Calculations based on the EXPLO 7 program show that most of detonation products of DPPE-1 are non-toxic and less toxic, with a mass percentage of I_2 of 51.7% (Table R3). To confirm the I_2 in the detonation product, (a) we filled the DPPE-1 in a pressure-resistant glass bottle and heated it in an oven to 200 °C. As a result, we heard a huge explosion and the bottle was completely broken (Figure R1a-b); (b) we filled DPPE-1 in a Teflon reactor and heated it in an oven to 200 °C. As a result, no explosion was heard and the detonation product stained the inner wall of the container purple, indicating the possible formation of I_2 (Figure R1c-d). Further tests showed that the aqueous solution of the purple substance was yellow and immediately formed a blue solution after mixing with starch (Figure R1e-g), confirming the presence of iodine. The content of I_2 based on the color comparison method is > 50%, which is consistent with the theoretical calculation. According to the above analysis, it is reasonable to think that IO_4^- is preferred to ClO_4^- to build primary explosives.

In response to your above question, we made a detailed explanation in the revised manuscript (See page 7, lines 24-33 and page 8, lines 1-44).

Table R1. Detonation performance of DPPE-1.

Items	DPPE-1	
Temperature (T) / K	173	298
Density (d) / g cm ⁻³	2.88	2.74
Heat of formation (ΔH) / kJ mol ⁻¹	-2928.80	-2928.80
Detonation velocity (D) / m s ⁻¹	5403	5156
Detonation pressure (P) / GPa	18.2	16.4

The above calculation was completed using the EXPLO7.0 program

Table R2. The standard electrode potential and Gibbs free energy.

Compounds	$(\text{H}_2\text{dabco})_2[\text{Na}(\text{NH}_4)(\text{IO}_4)_6]_n$	$(\text{H}_2\text{dabco})_2[\text{Na}(\text{NH}_4)(\text{ClO}_4)_6]$
Electrode reaction	$2\text{IO}_4^- + 16\text{H}^+ + 14\text{e}^- \rightarrow \text{I}_2 + 8\text{H}_2\text{O}$	$\text{ClO}_4^- + 8\text{H}^+ + 8\text{e}^- \rightarrow \text{Cl}^- + 4\text{H}_2\text{O}$
Standard electrode potential (E^θ) / V	1.314	1.389
Gibbs free energy (ΔG^θ) / kcal mol ⁻¹	-144.95	-201.98

E_0 , Refer to literature "G. Milazzo et al 1978 J. Electrochem. Soc. 125 261C";
 ΔG , obtained through theoretical calculations.

Figure R1. Experiment of detecting iodine (I_2) in explosive products.

Table R3. The detonation product of DPPE-1 calculated based on the EXPLO 7 program.

Detonation product	Mass percentage / %
H ₂ O	16.8177
C	5.3325
I₂	51.7077
CO ₂	8.7824
N ₂	4.6804
CH ₂ O ₂	5.0482
CO	1.0721
Na ₂ CO ₃	3.7457
NH ₃	0.3267
CH ₄	0.2729
HI	2.1273
H ₂	0.0206
C ₂ H ₆	0.0603
C ₂ H ₄	0.0032
HCN	0.0017
CH ₃ OH	0.0013
Total	100

Comment 3: *"Thermal stability: The thermal decomposition temperature of DPPE-1 is not outstanding, although it has similar thermal stability with the famous DDNP. But with the increasing safety requirements in the field of explosives, it was required that the thermal decomposition temperature of explosives should reach 180 degree. It is true that the current thermal stability has indeed reached the basic requirements for practical applications, but 161.3 degree is really not sufficient."*

Response: Thank you very much for your positive comments on our work. As you said, "the thermal decomposition temperature of DPPE-1 is not outstanding", but "It is true that the current thermal stability has indeed reached the basic requirements for practical applications". In fact, we are currently trying to develop a new periodate-based perovskite primary explosive with the purpose of ".....search for perovskite-type green primary explosives with thermal decomposition temperatures higher than 200 °C or even 250 °C" that we mentioned in the conclusion. It is of great academic and commercial value to achieve this goal, as I have done with ICM-103 research (*Nat. Commun.*, 2019, 10, 1339, four professional institutions have jointly carried out commercial application). In any case, DPPE-1 reported in this manuscript is of great significance for promoting the development of a perovskite primary explosive, which is what we are pursuing. We hope that you will appreciate our work. In response to your above question, we made a brief description in the revised manuscript (See page 5, lines 42-44 and page 6, line 1).

Response to Reviewer #2

Comment 1: "Congratulations again to the outstanding and impressive new perovskite compound. It is really simple and could be of potential use.

I totally recommend publication of this manuscript in the present journal. Some of the overdrawn formulations have been changed from the previous version. This is gratefully acknowledged."

Conclusion: The referee provided us with positive comments on our manuscript, e.g., "Congratulations again to the outstanding and impressive new perovskite compound" and "..... totally recommend publication of this manuscript in the present journal".

Comment 2: "I still have a strong problem with table 1 line mpc. The values given here for the other compounds are simply not correct. All of the references given here are not determining the mpc, they only give examples or amounts of a specific detonator setup. So either you have to determine the mpc of the other compounds in your setup, with all desired values such as pressure, grain size etc. or this line should be deleted. At least the values of the compared explosives must be deleted.

I do not want to say your perovskite is bad and lower than others but the authors should know that one cannot compare reference data using different setups and PETN and RDX charges for comparison. In our preliminary tests we achieved a similar initiation capability to LA which is quite nice.

To summarize, this work should be published and is of great interest to other researchers in high energy materials research"

Response: We really appreciate the reviewer's conclusion that "this work should be published and is of great interest to other researchers in high energy materials research". In addition, we thank the reviewer for pointing out our carelessness. According to the reviewer's suggestion, the line MPC in Table 1 has been deleted in the second revised manuscript (see Table 1). Upon careful examination, we found that the charge mass (50 mg) of K₂DNAT reported in the cited literature (*Angew. Chem. Int. Ed.* 2015, 54, 10299-10302) may not be the minimum primary charge (MPC). Considering that (1) K₂DNAT is extremely sensitive, and the detonation velocity and pressure are as high as 10011m/s and 52.2GPa respectively, it is extremely dangerous to carry out synthesis, particle size screening and ignition performance test, and (2) the data of test pressure and grain size of the other four initiating substances are missing, we decided to delete line MPC in Table 1 to avoid trouble caused by using different setups and secondary explosives (PETN and RDX).

Comment 3: "We have checked the synthesis, which worked quite fine. The analytic data largely fit to our observations. Our DTA was a bit lower for example. The methodologies used by the authors are appropriate and meet the desired standards. Good luck for the future. I know there are other Chinese research groups working on the exact same class of compounds."

Response: We appreciate you confirming our work through experiments and giving positive comments.

Response to Reviewer #3

Comment 1: "In this manuscript, Feng et al. claimed that they find a new energetic double perovskite (named DPPE-1) could be a promising primary explosive. In particular, they reported a very low minimum primary charge (MPC) of 5 mg for DPPE-1. The present findings seems very impressive, but in my opinion, some conclusions are suspicious. Moreover, in considering that some very close analogues of energetic perovskite class have been reported in refs. 24, 37, 41, and 45-49, the novelties of DPPE-1 (neither in structure nor the usage as primary explosive) is not high enough for publication on Nature Communication."

Response: We cannot agree with your above statement. (1) The statement that "*in my opinion, some conclusions are suspicious*" is not appropriate. We believe that making such comments should not be based on subjective speculation, but on solid experimental data. The synthetic method is very simple, and all the raw materials are commercially available, which makes it very easy for you to verify the present results. We would appreciate it if you could reproduce our results. We also believe this will help you make accurate judgments, just as Reviewer #2 did (*Comment 3*). (2) The statement that "*some very close analogues of energetic perovskite class have been reported in refs. 24, 37, 41, and 45-49*" doesn't hold water. We think that above view is a crucial factor for you to make the conclusion that "*the novelties of DPPE-1 is not high enough for publication on Nature Communication*". In our view, such a judgment is not rigorous. The DPPE-1 reported in our manuscript shows a double-perovskite structure, while other substances reported in refs. 24, 37, 41, and 45-49 show single-perovskite structure, which undoubtedly proves that DPPE-1 is different from known energetic perovskites. Although they share some components, it would be a mistake to classify them as very close analogues. In addition, growth in numbers usually implies new variations, possibilities, and functions. For single perovskites, there are theoretically N^3 ($N = 1,2,3,\dots,n$) molecular combinations, while for double perovskite, the number may be extended to N^4 ($N = 1,2,3,\dots,n$). Obviously, double-perovskite energetic materials have more possibilities. What reason do we have to think that its structure is not novel enough? Why isn't the increase in possibilities caused by structural changes an innovation? To take a similar example, it is well known that graphite, diamond, fullerene, graphene, carbon nanotubes and so on have exactly the same composition (C), but scientists did not believe that these new species were not an innovation. The main reason is that the slight changes in structure have led to many important discoveries, among which the discovery of fullerene and graphene has been awarded the Nobel Prize. The same principle applies to energetic perovskite.

Additionally, all known perovskite materials have no ignition function, while DPPE-1 has been proved to have excellent ignition performance. Isn't this innovation sufficient enough? To take a similar example, many new functions of the perovskite $\text{CH}_3\text{NH}_3\text{PbX}_3$ have been discovered by scientists in recent years. However, the complete consistency in chemical composition does not deny the innovation of these works, nor does it reject their publication in *Nature* and *Science*.

In response to your above question, we made a detailed description in the revised manuscript (See page 4 lines, 11-21 and page 7, lines 13-20). We sincerely hope that reviewer can reunderstand our innovations in structural construction and performance discovery.

Comment 2: "The space group is *Pa-3* in main text (and cif) but *Pcab* in Table S1, which one should be right? As the authors had not supplied any structure factors, the full set of tests cannot be run (see the first line in checkCIF/PLATON report). The authors should upload a completed cif file embedding HKL reflections and refinement details for fully checkcif. In addition, according to the EA result, the present single-crystal structure refinement may be wrong in the site occupancy of B-site components."

Response: Thank you very much for your detailed comments on our research, especially pointing out some serious errors in the manuscript, e.g., single crystal structure and elemental analysis, which we think will help improve the quality of our manuscript.

Crystal structure. The result of the initial refinement shows that DPPE-1 crystallized in the *orthorhombic* space group *Pcab*. However, it took us a long time to find out that this is in conflict with its obvious *cubic* system, and no energetic perovskites with *Pcab* space group have yet been reported (Table R4). Therefore, the crystal structure of DPPE-1 was refined for a second time, and it was finally identified as belonging to the *cubic* space group *Pa-3*. Unfortunately, this change was not corrected in time in the Supplementary Information, resulting in contradictory data. In response to this error, we have corrected it in the revised supporting document (see Table S1 lines 5-6). According to your request, we also uploaded the related refinement files, including completed cif files embedding HKL reflections and refinement details for fully checkcif (see DPPE-1 - 2nd refinement. cif/.fcf/.hkl). I think these documents will enable you to make a positive evaluation of our work.

Elemental analysis. Data errors in elemental analysis were identified as the result of instruments not being calibrated for a long time. After careful calibration, we redetermined the elemental composition of DPPE-1 (see Table S5), and the results showed that the contents of C, H and N were 10.14%, 2.19% and 4.89%, respectively. We have addressed this issue and you can see the corrected data on page 10, line 29 in the revised manuscript and on page S2 in the revised supplementary information.

Table R4. Crystal system and space group of some reported energetic perovskites.

Perovskite	CCDC Number	Formula	Crystal system	Space group
DAI-1 ^[1]	2164440	(H ₂ dabco)Na(IO ₄) ₃	cubic	Pa3
DAI-2 ^[1]	2164437	(H ₂ dabco)K(IO ₄) ₃	cubic	Pa3
DAI-3 ^[1]	2164438	(H ₂ dabco)Rb(IO ₄) ₃	cubic	Pa3
DAI-4 ^[1]	2164439	(H ₂ dabco)NH ₄ (IO ₄) ₃	cubic	Pa3
PAP-5 ^[2]	1984446	(H ₂ pz)[Ag(ClO ₄) ₃]	monoclinic	P2 ₁ /c
PAP-M5 ^[2]	1984448	(H ₂ mpz)[Ag(ClO ₄) ₃]	orthorhombic	Pnma
PAP-H5 ^[2]	1984449	(H ₂ hpz)[Ag(ClO ₄) ₃]	monoclinic	P2 ₁ /n
DAP-5 ^[2]	1984447	(H ₂ dabco)[Ag(ClO ₄) ₃]	cubic	Pa3
DAP-6 ^[3]	1978742	(H ₂ dabco)(NH ₃ OH)(ClO ₄) ₃	monoclinic	P2 ₁
DAP-7 ^[3]	1978743	(H ₂ dabco)(NH ₂ NH ₃)(ClO ₄) ₃	monoclinic	P2 ₁ /m
DAP-O4 ^[4]	1956808	(H ₂ dabco)[NH ₄ (ClO ₄) ₃]	cubic	Fm3c
PAP-4 ^[4]	1956805	(H ₂ pz)[NH ₄ (ClO ₄) ₃]	cubic	Fm3c
AP-M4 ^[4]	1956806	(H ₂ mpz)[NH ₄ (ClO ₄) ₃]	orthorhombic	Pnma
AP-H4 ^[4]	1956807	(H ₂ hpz)[NH ₄ (ClO ₄) ₃]	monoclinic	P2 ₁ /n

Perovskite	CCDC Number	Formula	Crystal system	Space group
DAP-M4 ^[4]	1956809	(H ₂ mdabco)[NH ₄ (ClO ₄) ₃]	monoclinic	P2 ₁
DAP-1 ^[5]	1528107	(H ₂ dabco)[Na(ClO ₄) ₃]	cubic	Pa3
DAP-2 ^[5]	1528106	(H ₂ dabco)[K(ClO ₄) ₃]	cubic	Pa3
DAP-3 ^[5]	1528109	(H ₂ dabco)[Rb(ClO ₄) ₃]	cubic	Pa3
DAP-4 ^[5]	1528108	(H ₂ dabco)[NH ₄ (ClO ₄) ₃]	cubic	Pa3
PAP-1 ^[6]	1872452	(H ₂ pz)[Na(ClO ₄) ₃]	monoclinic	P2 ₁ /c
DAP-O2 ^[6]	1860906	(H ₂ dabco-O)[K(ClO ₄) ₃]	cubic	Fm3c
DPPE-1	2173977	(C ₆ H ₁₄ N ₂) ₂ [Na(NH ₄ (IO ₄) ₆)]	cubic	Pa3

References

- [1] Sci China Mater., 2023, 66, 1641-1648
 [2] Inorg. Chem., 2022, 61, 4143-4149
 [3] Engineering, 2020, 6, 1013-1018
 [4] Cryst. Growth Des., 2020, 20, 1891-1897
 [5] Sci China Mater., 2018, 61, 1123-1128
 [6] CrystEngComm, 2018, 20, 7458-7463

Comment 3: "Page 9, line 2, "EA calculated for C₁₂H₃₂I₆N₅NaO₂₄ (1414.82 g mol⁻¹): C 10.19, H 2.28, N 4.95; Found: C 10.30, H 2.69, N 5.83." The experiment wt% value (5.83) for N element is significantly higher than the expected one (4.95). As disclosed in the ref. 41, NH₄⁺ could easily assemble into (H₂dabco)(NH₄)(IO₄)₃ (named DAI-4) with 75% yield, whereas sodium ion is relatively smaller for directly constructing (H₂dabco)[Na(IO₄)₃] (named DAI-1) from solution but giving (H₂dabco)[Na(H₄IO₆)₃] (DAI-X1) that could converted DAI-1 after dehydration at 80 °C. These facts in ref. 41, together with the mismatch in the present EA results, strongly imply that the sample used in the present manuscript is most-likely a mixture (main component should be DAI-4 with 5.96 wt% of N element, rather than the ideal DPPE-1 with 4.95 wt% of N element) or a solid solution (i.e., a larger amount of sodium sites are replaced by NH₄⁺ in the ideal DPPE-1) but not the declared pure DPPE-1. Regardless of the situations, it means that the obtained sample is more close to the known DAI-4, or the ideal DPPE-1 cannot be prepared as reliably as DAI-4, bringing troubles to subsequent applications. As all these relevant compounds have very close simulated PXRD patterns, the identify for the present sample should be very carefully (a pawley refinement maybe helpful in a certain degree)."

Response: Thank you very much for your detailed comments on our work. There is no doubt that the question you raise is valuable because it prompts us to re-examine and think about some fundamental issues. However, chemists should understand that it is untenable to determine substances solely based on nitrogen content, especially when the listed data may be incorrect.

From the nitrogen content of DPPE-1 (uncalibrated), the reported substance seems more likely to be a DAI-4-based mixture or solid solution. However, we also note that neither case explains why the contents of carbon and hydrogen of DPPE-1 (uncalibrated) are much higher than those of any substance listed in Table R5, including DAI-X1, DAI-1, and DAI-4. The most likely reason is that the data listed in the manuscript is wrong. Subsequently, we calibrated the instrument and remeasured the element composition (Table R6). The results show that the measured data

Table R5. Elements and contents of several perovskite materials.

Compounds	Elements and their contents / %		
	C	H	N
DAI-X1 (calculated)	8.81	3.2	3.42
DAI-1 (calculated)	10.15	1.99	3.95
DAI-4 (calculated)	10.22	2.57	5.96
DPPE-1 (calculated)	10.19	2.28	4.95
DPPE-1 (uncalibrated, listed on Page 9, line 2)	10.30	2.69	5.83
DPPE-1 (calibrated)	10.14	2.19	4.89

uncalibrated, testing without instrument calibration; calibrated, testing after instrument calibration

Table R6. Redetermined elemental analysis data.

Sample number	C/%	H/%	N/%
1#	10.16	2.22	4.86
2#	10.11	2.23	4.86
3#	10.12	2.17	4.91
4#	10.14	2.18	4.88
5#	10.17	2.16	4.92
Avg.	10.14	2.19	4.89

(DPPE-1 (calibrated)) are highly consistent with the theoretical calculation (DPPE-1 (calculated)). Therefore, the substance we reported is not DAI-4 but DPPE-1. We have addressed this issue and you can see the corrected data on page 10, line 29 in the revised manuscript and on page S2 in the revised supplementary information.

The single-crystal X-ray diffraction is considered to be the "Gold Standard" for structural determination. The crystal structure in the manuscript clearly confirms that the reported substance is DPPE-1 rather than DAI-4. Furthermore, if DPPE-1 cannot be prepared as reliably as DAI-4, this means that the acquisition of DPPE-1 is a low-probability event. If this is true, it would not explain why we always get DPPE-1 instead of DAI-4 or some other substance in the process of randomly selecting crystals for structural determination. Combined with the crystal structure, nuclear magnetic resonance and the calibrated elemental composition, we were able to conclusively confirm that the target substance was DPPE-1. In response to your above question, we have made a detailed supplementary explanation on the structure of DPPE-1 in the revised manuscript (See page 4, lines 7-11 and 19-21).

Comment 4: "For MPC test, the present lead-plate experiment is relatively preliminary. There are many factors highly affect MPC, such as the sample purity, crystalline form, crystal quality, crystal size and morphology. Therefore the reported MPC in different literature usually fluctuate within a certain range. In addition, the practical primary charge is not only determined by MIC, but also closely related to the detonation reliability under different conditions. For instance, different from the MPC of 30 mg for well-known LA reported in ref. 10, the MPC of 7 mg for LA is usually found in some textbooks, and the practical primary charge of 30 mg LA is usually needed in order to achieve stable detonation every time. Moreover, in order to compare the initiation ability of different initiating explosives, comparative experiments should be conducted under the same conditions. In addition, in considering that the present DPPE-1 sample may be a mixture or solid solution which do not easily and reliably prepared, I strongly suggest the author to study the potential as primary explosive for DAI-4."

Response: Thank you very much for your detailed comments on our work. However, the suggestion "to study the potential as primary explosive for DAI-4" indicates that you may not understand the importance of our research. The reasons are as follows: (1) The innovations of this research are that it expands the number of energetic perovskites through a new structure (from N^3 of single perovskites to N^4 of double perovskites, $N=1,2,3,\dots,n$) and discovers a new function of energetic perovskites, namely ignition performance. Energetic materials experts would be able to take note of these two points and immediately realize the great value of our research. Here, the purpose of MPC test is to confirm the desired ignition performance, it is not necessary to make a detailed analysis of various factors that affect MPC, such as purity, form, quality, size, and morphology. This is a protracted work; (2) It is well known that there is a considerable time span (ranging from a few years to a dozen years) from basic research, applied research, engineering technology to the final product. In this sense, it is highly impractical to try to solve all the issues related to DPPE-1 in a single research paper. Generally, the practical charge is considered in the process of initiating explosive devices development (depending on the application environment and product type), while what we are doing now is basic research that serves as an inspiration to other scientists and engineers. In other words, it is meaningless to discuss the practical charge of DPPE-1 in this paper; (3) The MPC of primary explosives is determined according to a predetermined standard, such as GJB5891-2006 adopted by scientists in China. In this standard, there are no requirements on the purity, crystalline form, crystal quality, crystal size and morphology of the sample. With the exception of K_2DNAT , the MPCs of the other primary explosives (e.g. DDNP, ICM-103, ANTPA, LA) were measured without taking into account the above influencing factors. (4) Considering that DPPE-1 is more novel than DAI-4 in both structure and performance, and that the misinterpretation raised about the structure of DPPE-1 have been proved to be incorrect, we do not accept your proposal to study DAI-4.

Comment 5: "The writing is not good and rigorous. Many author names are wrong in reference lists."

Response: The language, novelty and importance of the manuscript have been evaluated and polished by professional institutions. Even so, we will try our best to improve our writing.

文书标题 : Organic-Inorganic Perovskites as a New Platform for High-Performance Primary Explosives
服务类型 : 高级科学润色 [已完成] 字数: 4672字
专业信息 : 工程与技术科学 > 材料科学
优惠券 : MogoEdit202205212332571089638 (90折)
文件列表 :
支付方式 : 在线支付
查看详情 | 保密协议 | 填写调查问卷 获取9折优惠券

Figure R2. Order for professional evaluation.

Finally, we hope that you will appreciate the two important innovations involved in this study and make very positive comments accordingly.

Response to Reviewer #4

Comment 1: "The authors report a perovskite based promising Primary explosive. The work is interesting and the authors have done a systematic experimental study to address the energetic performance. However, I still feel that the Chemistry part of the manuscript is still not upto the level of Nature Communications. The major draw back of the paper is that there is no theory part to complement the experiments to make the story complete."

Conclusion: The referee provided us with positive comments on our manuscript, e.g., "*The work is interesting*" and "*the authors have done a systematic experimental study*". The referee thinks this work needs some additional theoretical explanation and also provides quiet a few constructive suggestions. These suggestions were extremely valuable and immensely helpful for revision and improvement of our paper. We would like to answer all the reviewer's questions. According to the reviewer's suggestion, we explained the mechanism in detail in the revised manuscript (see page 7, lines 24-33 and page 8, lines 1-16).

Comment 2: "The studied compound (perovskite) is already well established and this is not the first time reported in the literature. Too much emphasis on this is not a good idea."

Response: We agree with your statement of "*Too much emphasis on this is not a good idea*". However, readers' preferences are usually diametrically opposed. Some experts think that such an emphasis (e.g., first) is necessary, while others feel disgusted. Therefore, the authors are always in a dilemma. We thought that such emphasis is necessary. For example, if such an emphasis were removed from the manuscript, most reviewers would easily assume that this substance is no different from the energetic perovskite reported before. Unfortunately, our strategy seems to have failed to meet expectations. For example, we find that you think "*the studied compound (perovskite) is already well established and this is not the first time reported in the literature*". However, based on our knowledge, all reported energetic perovskites show a single perovskite framework, while double perovskite energetic materials have not been reported so far. The word "*first*" emphasizes both advances in new structure and, more importantly, our discovery in ignition function. In any case, we have corrected such a description in the revised manuscript according to your suggestion (See page 1, line 13).

*Comment 3: "I want the authors to compare the performance of the present compound with some other Azide such as NH_4N_3 , AgN_3 , CuN_6 etc. This will be very much useful for the readers. The authors should also consider Potassium 1,1'-dinitroamino-5,5'-bistetrazolate (K_2DNABT) for comparison. Please see the following paper {*J. Chem. Phys.* 143, 064508 (2015)}. I suggest the authors to include the above paper in the introduction part."*

Response: We agree with your suggestion. In the revised manuscript and supplementary information, we compare the performance of the present compound with those of the above-mentioned energetic substances (e.g. AgN_3 , CuN_6 and K_2DNABT , see page 7, lines 13-20). Ammonium azide (NH_4N_3) was not discussed because we did not find any data on its ignition performance in the possible references. The above paper has also been added to the introduction (see reference 11 on page 11, lines 28 -29).

Comment 4: "I would like to know the role of Halogen towards the energetic properties in this compound."

Response: This is the most valuable academic communication I think. Up to now, we have synthesized at least 50 energetic perovskite materials. In the next two years, the number of perovskite energetic materials in our laboratory will be increased more rapidly in order to provide the basis for screening primary explosives with better comprehensive properties. According to our available studies, halogens seem to play a decisive role in the initiation performance of energetic perovskites. For example, compared with ClO₄⁻-based materials, IO₄⁻-based materials are more likely to exhibit high sensitivity, rapid exothermic processes, and strong initiation performance, while BrO₄⁻-based materials are too unstable to be synthesized and used as energetic materials.

In the first revised manuscript, we explained that the ignition performance of DPPE-1 mainly depends on the strong oxidation ability of IO₄⁻. However, such an oversimplified explanation does not satisfy the reviewers. Here, we calculated the heat of formation of DPPE-1, and evaluated the detonation performance of DPPE-1 by using program EXPLO 7.0 (see Table R7). The results show that its detonation velocity (D) and detonation pressure (P) are < 5500 m/s and <18.5GPa, respectively, which are lower than those of most primary explosives, indicating that the impressive initiation performance of DPPE-1 is hard to be well explained from the energy level. So, we turned our attention to oxidation capacity. According to the Tables of Standard Electrode Potentials (*G. Milazzo et al 1978 J. Electrochem. Soc. 125 261C*), the standard electrode potential (E_0) of IO₄⁻ is 1.314V, while that of ClO₄⁻ is 1.389V. Obviously, the oxidation of ClO₄⁻ is stronger than that of IO₄⁻, which shows that the good initiation performance of DPPE-1 may be related to some unknown influencing factors besides the strong oxidation of IO₄⁻. We think that molecular stability plays a crucial role, so we compare the Gibbs Free Energy (ΔG) of (H₂dabco)₂[Na(NH₄)(IO₄)₆]_n and (H₂dabco)₂[Na(NH₄)(ClO₄)₆]_n (see Table R8). The results show that the ΔG of (H₂dabco)₂[Na(NH₄)(IO₄)₆] ($\Delta G = -144.95$ kcal/mol) is higher than that of (H₂dabco)₂[Na(NH₄)(ClO₄)₆] ($\Delta G = -201.98$ kcal/mol), that is, the ClO₄⁻ in (H₂dabco)₂[Na(NH₄)(ClO₄)₆] has a good stabilizing effect, while the material based on IO₄⁻ is relatively unstable, which may help us to understand the high mechanical sensitivity and strong initiation performance of DPPE-1. Such in-depth understanding is inseparable from the continuous support of reviewers, and we are very grateful for your kindness and patience.

In response to your above question, we have made a detailed supplementary explanation on the role of halogen towards the energetic properties in the revised manuscript (See page 7, lines 27-33 and page 8, lines 1-16).

Table R7. Detonation performance of DPPE-1.

Items	DPPE-1	
Temperature (T) / K	173	298
Density (d) / g cm ⁻³	2.88	2.74
Heat of formation (ΔH) / kJ mol ⁻¹	-2928.80	-2928.80
Detonation velocity (D) / m s ⁻¹	5403	5156
Detonation pressure (P) / GPa	18.2	16.4

The above calculation is completed through the EXPLO7.0 program

Table R8. The standard electrode potential and Gibbs free energy.

Compounds	(H ₂ dabco) ₂ [Na(NH ₄)(IO ₄) ₆]	(H ₂ dabco) ₂ [Na(NH ₄)(ClO ₄) ₆]
Electrode reaction	2IO ₄ ⁻ + 16H ⁺ + 14e ⁻ ⇌ I ₂ + 8H ₂ O	ClO ₄ ⁻ + 8H ⁺ + 8e ⁻ ⇌ Cl ⁻ + 4H ₂ O
Standard electrode potential (E ₀) / V	1.314	1.389
Gibbs free energy (ΔG) / kcal mol ⁻¹	-144.95	-201.98

*E*₀, Refer to literature "G. Milazzo et al 1978 J. Electrochem. Soc. 125 261C";

ΔG, obtained through theoretical calculations.

Comment 5: "The Halogens are toxic and I would like to know how the present compound will satisfy the green energetic materials criteria."

Response: The following four references have listed the criteria that green primary explosives need to meet:

- (1) Z. Anorg. Allg. Chem. 2014, 640, 1309-1313;
- (2) Angew. Chem. Int. Ed. 2014, 53, 8172-8175;
- (3) Oyler, K. D. Green Primary Explosives. John Wiley & Sons, Ltd. 2014;
- (4) Proc. Natl. Acad. Sci. USA 2006, 103, 5409-5412.

According to the statements in these documents, the criteria of green primary explosives can be summarized as follows:

- (1) must be safe to handle and possess a rapid deflagration to detonation transition;
- (2) must be thermally stable to greater than 150 °C;
- (3) should possess high detonation performance and sensitivity;
- (4) should have long term chemical stability;
- (5) should not contain toxic heavy metals, perchlorate or other known toxins;
- (6) easy and safe to synthesize and affordable;
- (7) insensitivity to light;
- (8) stability upon storage for long periods of time.

Obviously, DPPE-1 meets all the above criteria. The focus is on how to explain the toxicity of halogen. In fact, we must be aware that all reported primary explosives are extremely toxic to organisms. So why are some of them identified as green primary explosives? Looking back on the development history of primary explosives, we will find that green primary explosives were introduced to reduce the heavy metal toxicity of lead azide (LA) and lead styphnate (LS). Theoretically, all substances without heavy metals can be regarded as green primary explosives. This is the main reason why some substances are classified as green primary explosives (Oyler, K. D. Green Primary Explosives. John Wiley & Sons, Ltd. 2014) even though they contain extremely high toxic components, including silver azide (SA), nickel hydrazine nitrate (NHN), copper(I) 5-nitrotetrazolate (DBX-1), bis-(5-nitrotetrazole)tetraamine cobalt(III) perchlorate (BNCP), 2-diazo-4,6-dinitrophenol (DDNP), cyanuric triazide (CTA), potassium 4,6-dinitro-7-hydroxybenzofuroxan (KDNP) and potassium 4,6-dinitrobenzofuroxan (KDNBF). As for IO₄⁻, its toxicity is much lower than that of the components such as hydrazine, azide, 5-nitrotetrazolate, nitrophenol, benzofuroxan and cyanuric triazide in the above-mentioned green primary explosives. In addition, most of its

detonation products are non-toxic and low toxic (see Table R9), and the main solid product I₂ has bactericidal and disinfection function, which makes it a possible promising energetic biocidal agent (see Figure R3. *Sci China Mater.*, 2023, 66, 1641-1648). To sum up, DPPE-1 satisfy the green energetic materials criteria. In the revised manuscript, we explain this in detail (See page 8, lines 17-44).

Table R9. The detonation products of DPPE-1 calculated based on the EXPLO 7 program.

Detonation product	Mass percentage / %
H ₂ O	16.8177
C	5.3325
I₂	51.7077
CO ₂	8.7824
N ₂	4.6804
CH ₂ O ₂	5.0482
CO	1.0721
Na ₂ CO ₃	3.7457
NH ₃	0.3267
CH ₄	0.2729
HI	2.1273
H ₂	0.0206
C ₂ H ₆	0.0603
C ₂ H ₄	0.0032
HCN	0.0017
CH ₃ OH	0.0013
Total	100

In response to the comment 2 of reviewer #1, we experimentally confirmed the presence of I₂ in solid products.

SCIENCE CHINA Materials
ARTICLES

mater.scichina.com link.springer.com
Published online 2 December 2022 | <https://doi.org/10.1007/s40843-022-2257-6>
Sci China Mater 2023, 66(4): 1641-1648

 CrossMark
click for updates

Periodate-based molecular perovskites as promising energetic biocidal agents

Zhi-Hong Yu, De-Xuan Liu, Yu-Yi Ling, Xiao-Xian Chen, Yu Shang, Shao-Li Chen, Zi-Ming Ye, Wei-Xiong Zhang* and Xiao-Ming Chen

Figure R3. Periodate-based perovskites as promising energetic biocidal agents.

Ref. : Sci China Mater., 2023, 66, 1641-1648

Comment 6: "Can this compound be a perfect replacement for PbN₆ if so how superior it is when compared to K₂DNABT. I want the authors to compare the same. Also K₂BDAF should also be compared."

Response: Frankly speaking, DPPE-1 can't be a perfect replacement for PbN₆ (LA) unless its thermal decomposition temperature is higher than 200°C. Fortunately, it is very suitable as a substitute for DDNP at present. In the future, our focus will be on finding IO₄⁻-based perovskite primary explosive with thermal decomposition temperature higher than 200 °C or even 250 °C, which will be research with great commercial value. As for K₂DNABT and K₂BDAF, they may be high-performance primary explosives, but the lengthy manufacturing process makes them almost impossible to replace LA. Compared with these two potassium-based primary explosives, DPPE-1 has obvious advantages in crucial requirements, such as extremely simple and rapid synthesis, relatively high production safety and ultra-high initiation performance. More importantly, these advantages make it very convenient for readers to synthesize and then study how to improve its comprehensive performance.

According to the reviewer's suggestion, we have added some statements related to two potassium-based primary explosives in the revised manuscript (see page 7, line 13-20), which we believe is very valuable.

Comment 7: "As of now the comparison is not adequate for the energetic properties and I suggest the authors to revise the manuscript accordingly. Also the Hydrogen bonding part should be well discussed. The authors just present without much of discussion. I would like to review the paper again."

Response: Thank you very much for your constructive comments. According to your suggestion, we have made a detailed statement on the energetic properties and hydrogen bonding in the revised manuscript (see page 5, lines 13-21).

Reviewers' Comments:

Reviewer #1:

Remarks to the Author:

After carefully reading the comments of the other two reviewers as well as the authors' responses, I agree with the third reviewer's viewpoint and still think that this paper is not suitable for publication in Nature Communications, and its main problem is that the structure presented in this paper is really too similar with the compounds reported. I have to cite the relevant literatures once again to support my viewpoint.

- 1) Small 2023, DOI: 10.1002/smll.202302631
- 2) Science China Materials, 2022, DOI: 10.1007/s40843-022-2257-6
- 3) Inorganic Chemistry, 2022, DOI: 10.1021/acs.inorgchem.1c03958
- 4) Engineering, 2020, DOI: 10.1016/j.eng.2020.05.018
- 5) Crystal Growth Design, 2020, DOI: 10.1021/acs.cgd.9b01592
- 6) Science China Materials, 2018, DOI: 10.1007/s40843-017-9219-9

In addition to the very similar structure, its potential application as primary explosive has also been scrutinized, see literature "Small 2023, 2302631, DOI: 10.1002/smll.202302631". In my opinion, this published Small paper is superior to the submitted manuscript, in terms of the structural diversity, experimental integrity and mechanistic explorations. It can also be found that the structures and compositions of these materials are very similar, all employing H₂DABCO and IO₄⁻, further suggesting that the structure in this manuscript is not novel.

Reviewer #2:

Remarks to the Author:

Wow, this is an impressive Letter to Reviewer. The authors put a lot of efforts in further improving this manuscript and to explain the curiosity of this class of compounds.

Thanks for deleting the MPC.

I hope this significant work will be published soon in Nature Comm.

Reviewer #3:

Remarks to the Author:

I agree the most of the responses from the authors, including those concerning the wrong EA results and the structure refinement. However, I am afraid that two main novelties of DPPE-1, an idea from a single perovskite energetic crystal to a double perovskite energetic crystal and an application using DPPE-1 as primary explosive, are still not insufficient to be published on Nat. Commun. In particular, recently I found out that a very similar work using periodate-based perovskite energetic materials (i.e., DAI-1/2/3/4 reported in ref. 42 by a group from Sun Yat-Sen university) to be primary explosives was reported in Small (2023, DOI: 10.1002/smll.202302631) by a group from Nanjing University of Science and Technology.

Reviewer #4:

Remarks to the Author:

The authors have taken enough care and revised the manuscript by taking all the suggestions/comments given by the referees. I want the authors to check all the references carefully. I do see mistakes in the title, page number etc. For example the title of Ref. 11 in the revised manuscript is incomplete. I will be happy to recommend the paper for possible publication in Nature Communications once the authors check all the references.

A Point by Point Response to the Reviewer's Comments

Response to Reviewer #1

Comment: After carefully reading the comments of the other two reviewers as well as the authors' responses, I agree with the third reviewer's viewpoint and still think that this paper is not suitable for publication in Nature Communications, and its main problem is that the structure presented in this paper is really too similar with the compounds reported. I have to cite the relevant literatures once again to support my viewpoint.

- 1) *Small* 2023, DOI: 10.1002/sml.202302631
- 2) *Science China Materials*, 2022, DOI: 10.1007/s40843-022-2257-6
- 3) *Inorganic Chemistry*, 2022, DOI: 10.1021/acs.inorgchem.1c03958
- 4) *Engineering*, 2020, DOI: 10.1016/j.eng.2020.05.018
- 5) *Crystal Growth Design*, 2020, DOI: 10.1021/acs.cgd.9b01592
- 6) *Science China Materials*, 2018, DOI: 10.1007/s40843-017-9219-9

In addition to the very similar structure, its potential application as primary explosive has also been scrutinized, see literature "Small 2023, 2302631, DOI: 10.1002/sml.202302631". In my opinion, this published Small paper is superior to the submitted manuscript, in terms of the structural diversity, experimental integrity and mechanistic explorations. It can also be found that the structures and compositions of these materials are very similar, all employing H₂DABCO and IO₄⁻, further suggesting that the structure in this manuscript is not novel.

Response: The reviewer has discussed the issue of structural novelty with us for three times. The earliest one occurred when the manuscript was submitted to *Nature* (see Figure R1). Up to now, we are still confused as to why the reviewer has consistently denied the significant structural differences between energetic single perovskites and energetic double perovskites. The similarity in composition cannot stifle innovation in structure. Nevertheless, we still appreciate the efforts made by the reviewer to improve the quality of our manuscript. In addition, we have cited all relevant references mentioned by the reviewer (as shown above) in our previous manuscript and conducted a detailed examination of the relevant references. However, as of the time we submitted to *Nature* and *Nature Communications*, we did not find any reports on energetic double-perovskite structures and perovskite primary explosives.

27th December 2022

Nature

A Promising Perovskite Primary Explosive

2022-12-20560

Contributing Author

**Manuscript
Transferred**

Figure R1. Record of submission to Nature.

As shown in Figure R2, the latest study on perovskite primary explosives was submitted to *Small* on March 2, 2023, which was later than our earliest submission to *Nature* on December 27, 2022. Clearly, we launched our research earlier.

It is even more noteworthy that: (1) DPPE-1, as reported by us, is still the only energetic double perovskite to date; (2) The synthesis method we reported is carried out at room temperature, and the raw materials used are relatively safe inorganic salts. On the contrary, the synthesis of single-

perovskite energetic materials reported in the paper published in *Small* needs to be carried out under higher temperature (65 °C) and strong oxidizing acid (H₅IO₆), which is uneconomical and dangerous. As shown in Figure R3. (3) The TDPIs series of substances reported in the paper published in *Small* still lag behind DPPE-1 in terms of initiation capacity (5mg of DPPE-1 vs 10mg of TDPI-1), which actually proves that the ignition performance of double-perovskite energetic materials is better than that of single perovskite energetic materials. In other words, TDPIs highlight the significant advantages of DPPE-1, rather than "this published *Small* paper is superior to the submitted manuscript".

and [H₂DABCO][NH₄(IO₄)₃] were synthesized as follows: Similar procedure as that for [H₂DABCO][Na(IO₄)₃] was followed, except that 0.25 g (3.3 mmol) of KCl and 0.18 g (3.3 mmol) of NH₄Cl were used, respectively.

Synthesis of [H₂DABCO][M(ClO₄)₃] (DAP-1, 2, and 4; M=Na⁺, K⁺, and NH₄⁺): For [H₂DABCO][Na(ClO₄)₃], equimolar amounts of NaCl (0.30 mg, 5 mmol) and C₆H₁₂N₂ (0.56 g) were dissolved in 10 mL of deionized water in a salinized glass vial. 2.15 g (15 mmol) HClO₄ was added into 10 mL of deionized water and heated to 60 °C. Then, the mixture solution was added dropwise into the solution of HClO₄. After the addition, the mixture was stirred at 65 °C for 20 min. After that, the white precipitates were filtered, washed with ethanol, and dried in the oven at 50 °C for 6 h. [H₂DABCO][K(ClO₄)₃] and [H₂DABCO][NH₄(ClO₄)₃] were synthesized with similar procedure as that for [H₂DABCO][Na(ClO₄)₃], except that 0.38 g (5 mmol) of KCl and 0.27 g (5 mmol) of NH₄Cl were used, respectively.

Measurements: Powder X-ray diffraction (XRD) pattern was acquired on a Bruker D8-Advance diffractometer using a Cu source scanned from 5°–80° with a step size of 0.02°. The crystal morphology was collected by scanning electron microscope (SEM, JSM-IT500HR, Japan) with a primary electron energy of 3–30 keV, a current of ≈1 nA, and a spot size of 1.5. TG/DSC were performed simultaneously to determine the thermal reaction of the sample at the rate of 10 k min⁻¹ under nitrogen atmosphere in a METTLER-TOLEDO TGA/DSC 3+. The combustion heat of the sample

Keywords

detonation performance, energetic materials, initiation capability, perovskite, thermal decomposition

Received: March 28, 2023
Revised: June 5, 2023
Published online:

[1] a) M. Deng, Y. Feng, W. Zhang, X. Qi, Q. Zhang, *Nat. Commun.* **2019**, *10*, 1339; b) C. He, J. M. Shreeve, *Angew. Chem., Int. Ed.* **2016**, *55*, 772.

[2] a) Y. Feng, S. Chen, Z. Li, T. Zhang, *Chem. Eng. J.* **2022**, *429*, 132186; b) T. W. Myers, J. A. Bjorgaard, K. E. Brown, D. E. Chavez, S. K. Hanson, R. J. Scharff, S. Tretiak, J. M. Veauthier, *J. Am. Chem. Soc.* **2016**, *138*, 4685.

[3] a) Q. Wang, X. Feng, S. Wang, N. Song, Y. Chen, W. Tong, Y. Han, L. Yang, B. Wang, *Adv. Mater.* **2016**, *28*, 5837; b) M. H. H. Wurzenberger, M. Lommel, M. S. Gruhne, N. Szimhardt, J. Stierstorfer, *Angew. Chem., Int. Ed.* **2020**, *59*, 12367.

Small **2023**, 2302631

2302631 (11 of 12)

© 2023 Wiley-VCH GmbH

ADVANCED SCIENCE NEWS
www.advancedsciencenews.com

NANO · MICRO
small
www.small-journal.com

Figure R2. Record of submission to *Small*.

Synthesis of [H₂DABCO][M(IO₄)₃] (TDPI-1, 2, and 4; M=Na⁺, K⁺, and NH₄⁺): [H₂DABCO][Na(IO₄)₃], as a typical example, was synthesized as follow steps. First, 2.28 g (10 mmol) H₅IO₆ was dissolved in 15.0 mL of deionized water and heated to 65 °C under vigorous stirring, forming a clear solution in a salinized glass vial. Equimolar amounts TEDA (0.37 g, 3.3 mmol) and NaCl (0.19 g, 3.3 mmol) were dissolved in 5.0 mL of deionized water. Then, the mixture solution was added dropwise into the solution of H₅IO₆. After the addition, the mixture was stirred at 65 °C for 20 min. After that, the white precipitates were filtered, washed with ethanol, and dried in the air oven at 50 °C for 6 h. [H₂DABCO][K(IO₄)₃] and [H₂DABCO][NH₄(IO₄)₃] were synthesized as follows: Similar procedure as that for [H₂DABCO][Na(IO₄)₃] was followed, except that 0.25 g (3.3 mmol) of KCl and 0.18 g (3.3 mmol) of NH₄Cl were used, respectively.

Synthesis of DPPE-1
Dabconium dihydrochloride (0.37 g, 2 mmol) and ammonium chloride (0.0535 g, 1 mmol) were dissolved in 5 mL water by vigorous agitation (600 r·min⁻¹) at room temperature. Subsequent addition of 8 mL sodium metaperiodate (NaIO₄, 1.28 g, 6 mmol) solution into the above mixture resulted in the immediate precipitation of a white solid and the reaction solution became clear and colorless within 2–3 seconds. The resulting precipitate was filtered, washed with an ice/water mixture (2 × 3 mL), and then sequentially dried under sunlight and vacuum to yield the target compound DPPE-1 as a colorless solid. Yield: 1.02 g, 72.1%. DSC (5 °C·min⁻¹, °C): 161.3 (dec.); IR (KBr pellet, cm⁻¹): ν 3122 (m), 3034 (w), 1475 (m), 1419 (s), 1328 (w), 1214 (m), 1056 (s), 830 (s). ¹H NMR (600 MHz, DMSO-d₆, 25 °C): δ = 7.06 ppm (1H, NH), 3.36 (2H, CH₂); ¹³C NMR (600 MHz, DMSO-d₆, 25 °C): δ = 43.90 ppm; EA calculated for C₁₂H₂₁N₅NaO₂₁ (1414.82 g·mol⁻¹): C 10.19, H 2.28, N 4.95; Found: C 10.14, H 2.19, N 4.89.

Figure R3. Comparison of synthesis methods.

Once again, we thank you for your great efforts in reviewing this manuscript.

Response to Reviewer #2

Comment: Wow, this is an impressive Letter to Reviewer. The authors put a lot of efforts in further improving this manuscript and to explain the curiosity of this class of compounds.

Th aks for deleting the MPC.

O hope this significant work will be published soon in Nature Comm.

Response: We really appreciate your positive comments on our research. At the same time, we thank you for your great efforts to review and improve the quality of this manuscript.

Response to Reviewer #3

Comment: I agree the most of the responses from the authors, including those concerning the wrong EA results and the structure refinement. However, I am afraid that two main novelties of DPPE-1, an idea from a single perovskite energetic crystal to a double perovskite energetic crystal and an application using DPPE-1 as primary explosive, are still not insufficient to be published on *Nat. Commun.* In particular, recently I found out that a very similar work using periodate-based perovskite energetic materials (i.e., DAI-1/2/3/4 reported in ref. 42 by a group from Sun Yat-Sen university) to be primary explosives was reported in *Small* (2023, DOI: 10.1002/smll.202302631) by a group from Nanjing University of Science and Technology..

Response: We have carefully checked the references related to perovskite energetic materials.. However, as of the time we submitted to *Nature* and *Nature Communications*, we did not find any reports on energetic double-perovskite structures and perovskite primary explosives.

As we replied to the reviewer #1, (1) the latest study on perovskite primary explosives was submitted to *Small* on March 2, 2023, which was later than our earliest submission to *Nature* on December 27, 2022. Clearly, we launched our research earlier. (2) DPPE-1, as reported by us, is still the only energetic double perovskite to date; (3) The synthesis method we reported is carried out at room temperature, and the raw materials used are relatively safe inorganic salts. On the contrary, the synthesis of single-perovskite energetic materials reported in the paper published in *Small* needs to be carried out under higher temperature (65°C) and strong oxidizing acid (H₅IO₆), which is uneconomical and dangerous. As shown in Figure R3. (4) The TDPIs series of substances reported in the paper published in *Small* still lag behind DPPE-1 in terms of initiation capacity (5mg of DPPE-1 vs 10mg of TDPI-1), which actually proves that the ignition performance of double-perovskite energetic materials is better than that of single perovskite energetic materials.

Response to Reviewer #4

***Comment:** The authors have taken enough care and revised the manuscript by taking all the suggestions/comments given by the referees. I want the authors to check all the references carefully. I do see mistakes in the title, page number etc. For example the title of Ref. 11 in the revised manuscript is incomplete. I will be happy to recommend the paper for possible publication in Nature Communications once the authors check all the references.*

Response: We really appreciate your positive comments on our research. At the same time, we thank you for your great efforts to review and improve the quality of this manuscript. We have carefully checked and revised the references according to your suggestion.